evolution, genomics

asexual reproduction, parthenogenesis, sex-biased genes, sexual conflict, automixis, crustaceans

**Authors for correspondence:**
Ann Kathrin Huylmans
e-mail: a.huylmans@ist.ac.at
Beatriz Vicoso
e-mail: beatriz.vicoso@ist.ac.at

# Transitions to asexuality and evolution of gene expression in *Artemia* brine shrimp

Ann Kathrin Huylmans[1], Ariana Macon[1], Francisco Hontoria[2] and Beatriz Vicoso[1]

[1]Institute of Science and Technology Austria, Am Campus 1, Klosterneuburg 3400, Austria
[2]Instituto de Acuicultura de Torre de la Sal (IATS-CSIC), 12595 Ribera de Cabanes, Castellón, Spain

AKH, 0000-0001-8871-4961; FH, 0000-0003-2466-1375; BV, 0000-0002-4579-8306

While sexual reproduction is widespread among many taxa, asexual lineages have repeatedly evolved from sexual ancestors. Despite extensive research on the evolution of sex, it is still unclear whether this switch represents a major transition requiring major molecular reorganization, and how convergent the changes involved are. In this study, we investigated the phylogenetic relationship and patterns of gene expression of sexual and asexual lineages of Eurasian *Artemia* brine shrimp, to assess how gene expression patterns are affected by the transition to asexuality. We find only a few genes that are consistently associated with the evolution of asexuality, suggesting that this shift may not require an extensive overhauling of the meiotic machinery. While genes with sex-biased expression have high rates of expression divergence within Eurasian *Artemia*, neither female- nor male-biased genes appear to show unusual evolutionary patterns after sexuality is lost, contrary to theoretical expectations.

## 1. Introduction

Sex is nearly ubiquitous in animals, despite the costs associated with sexual reproduction. Asexual lineages typically appear at the tip of phylogenies, reflecting their short existence [1–3]. Sex is, therefore, presumed to confer an evolutionary advantage, primarily through the action of recombination in increasing genetic diversity, which in turn makes natural selection more effective [4–9]. According to theory [8,10,11], asexual species should have lower genetic diversity, reduced levels of adaptation, larger numbers of transposable elements and generally lower fitness than their sexual relatives. Groups where both sexual and asexual species or populations can be found have been used to test these hypotheses [12–17]. Fitness levels are hard to estimate, and most studies have instead focused on molecular predictions. Surprisingly, asexual populations often have similar diversity, rates of adaptation and numbers of transposable elements as their sexual relatives [11,13,18–22]. It is, therefore, still unclear whether shifts to asexuality consistently bring the changes in fitness that are predicted by theory.

The molecular mechanisms underlying asexuality are largely unknown, and the extent to which the switch to asexuality itself is associated with large shifts in expression is still under debate. Comparisons of gene expression have been used to investigate both the molecular basis and the consequences of asexuality [16,23,24], and detected many hundreds of genes differentially expressed between sexual and asexual morphs/lineages. Similarly, Duncan *et al.* [25] found evidence of an entirely different developmental programme underlying the asexual part of the life cycle of the pea aphid. On the other hand, single loci often seem to control the shift between sexual and asexual states [26–28]. Whether this simple genetic architecture translates into large or subtle changes in gene expression is still unknown.

How the peculiar selective pressures that asexual lineages face influence the evolution of their patterns of expression is also unclear. If the efficacy of selection is simply reduced, gene expression should progressively drift away from its optimum. In this case, sexual and asexual lineages may diverge quickly in their expression patterns, but in a largely random manner. However, there is evidence that more complex differences exist between the selective pressures acting on sexual and asexual populations. van der Kooi & Schwander [29] reviewed the decay of sexual traits after the evolution of asexuality in animals and showed that male traits, which are presumably under relaxed purifying selection, do not decay as quickly as sexual female traits (such as ones required for mate finding), suggesting that the latter are costly and are thus actively selected against. The changes in meiosis involved in the transition to asexuality also lead to convergent evolution at the level of gene sequences and expression, indicating a further role for selection [16,17].

Finally, asexual lineages may have a selective advantage relative to sexual lineages, as they are relieved from the 'intralocus sexual conflict' that occurs when alleles or genes are beneficial to one sex but deleterious to the other [11,30]. When such conflict is prevalent, sexual species experience a 'gender load', as neither sex is able to reach its own optimum. In asexual lineages, selection occurs exclusively in females, overcoming this limitation. Parker *et al.* [23] proposed that asexual females should, therefore, be able to invest more in female functions, which they assessed by testing if genes expressed primarily in females of stick insects increased their expression after switches to asexuality. On the contrary, asexual females had lower levels of expression for genes that are female-biased in sexual lineages, and higher levels for male-biased ones (they had 'masculinized' patterns of gene expression), consistent with a change in female trait optima in the asexual lineages and the decay of sexual traits. It is still unclear if this masculinization reflects a general trend of asexual species, and, if so, whether decay of female functions is the main force driving it.

Eurasian brine shrimp of the genus *Artemia* is a promising model for testing these theories. The clade consists of multiple sexual species as well as several parthenogenetic asexual lineages. The ancestral state of the group is sexual reproduction, with males and females showing extensive morphological dimorphism [31]. Asexual lineages are thought to have arisen multiple times as a result of 'contagious parthenogenesis': asexual females occasionally produce males that can fertilize females of closely related sexual populations, thereby giving rise to new asexual lineages [32–36]. However, the relationship between sexual and asexual lineages, and how diverged they are, are still questions under debate, which have mostly been investigated using a small number of mitochondrial and nuclear sequences [32,34,37]. How different these species and populations are at the gene expression level, and whether the asexual lineages share a single reproductive programme (as expected under contagious parthenogenesis), is still unknown. Here we describe a large RNA-sequencing dataset for the three sexual species (*Artemia sinica*, *Artemia urmiana sexual* and *Artemia* sp. *Kazakhstan*) as well as three diploid asexual *Artemia parthenogenetica* lineages (*Artemia parthenogenetica Aibi Lake*, *Artemia parthenogenetica Atanasovsko* and *Artemia parthenogenetica urmiana*, referred to, respectively, as Aaib, Aata and Aurm in the figures) of Eurasian *Artemia* (see Methods and the electronic supplementary material for details on their origin). We investigate the relationship between these using thousands of transcript sequences as well as patterns of genetic diversity. We characterize patterns of expression in males and females of sexual species, and compare them to expression patterns found in asexual females, allowing us to test whether a core set of genes changes consistently with the evolution of asexuality and whether we can detect a feminization in expression patterns in asexual females, consistent with a release from sexual antagonism.

## 2. Results

### (a) Phylogeny and evolution of asexuality

The phylogenetic relationship between the six Eurasian *Artemia* lineages used in this analysis was first established based only on mitochondrial data and two nuclear genes [33], which had suggested the existence of independent asexual lineages, including one more closely related to *A.* sp. *Kazakhstan* and another to *A. urmiana sexual*. A recent study using more detailed mitochondrial and nuclear markers found that all diploid asexual lineages are more closely related to *A.* sp. *Kazakhstan*, and probably originated through a complex demographic scenario involving a single origin of asexuality followed by backcrossing to *A.* sp. *Kazakhstan* [37]. To examine these relationships with genome-wide data, we sequenced RNA from multiple tissues for each lineage (electronic supplementary material, table S2), and assembled high-quality transcriptomes for each of them (electronic supplementary material, figure S1). The American species *Artemia franciscana* (Afra) [36] was used as the outgroup.

Protein sequences were inferred from each transcriptome with EviGene [38] (see Methods). Orthologous proteins were obtained by finding reciprocal best hits between all seven proteomes, then these 8389 sequences were aligned, concatenated and used for maximum-likelihood (ML) phylogenetic analysis with phyML. The resulting tree is in figure 1*a*. A similar topology was obtained with a Bayesian approach using BEAST2 (not shown). All sampled asexual lineages cluster with *A.* sp. *Kazakhstan*, with *A. urmiana sexual* being an outgroup to the cluster, consistent with [37]. These lineages (grey rectangle, figure 1*a*) are very closely related, and the lengths of the branches separating them are almost zero. We called genetic variants in each population and estimated pairwise Fst (figure 1*a*). The Fst analysis shows little genetic differentiation within the *A.* sp. *Kazakhstan*/asexuals clade, consistent with new strains of asexual *Artemia* having arisen repeatedly from an *A.* sp. *Kazakhstan*-related lineage in a very short amount of time [39].

The close relationship between *A.* sp. *Kazakhstan* and the asexual lineages is also reflected in the gene expression data. We profiled head and gonad expression, two sexually dimorphic organs (as males have modified antennae used for grasping females), from all lineages. The principal component analysis (PCA) of expression shows that head samples cluster primarily by species (figure 1*b*). In gonads, males cluster by species, but among females, only *A. sinica* forms a separate cluster. *A. urmiana sexual* and *A.* sp. *Kazakhstan* females overlap with asexuals, confirming the close relationship between asexual and sexual lineages. The PCA for the thorax shows the same pattern as heads, with *A.* sp. *Kazakhstan* clustering with the asexuals independent of sex

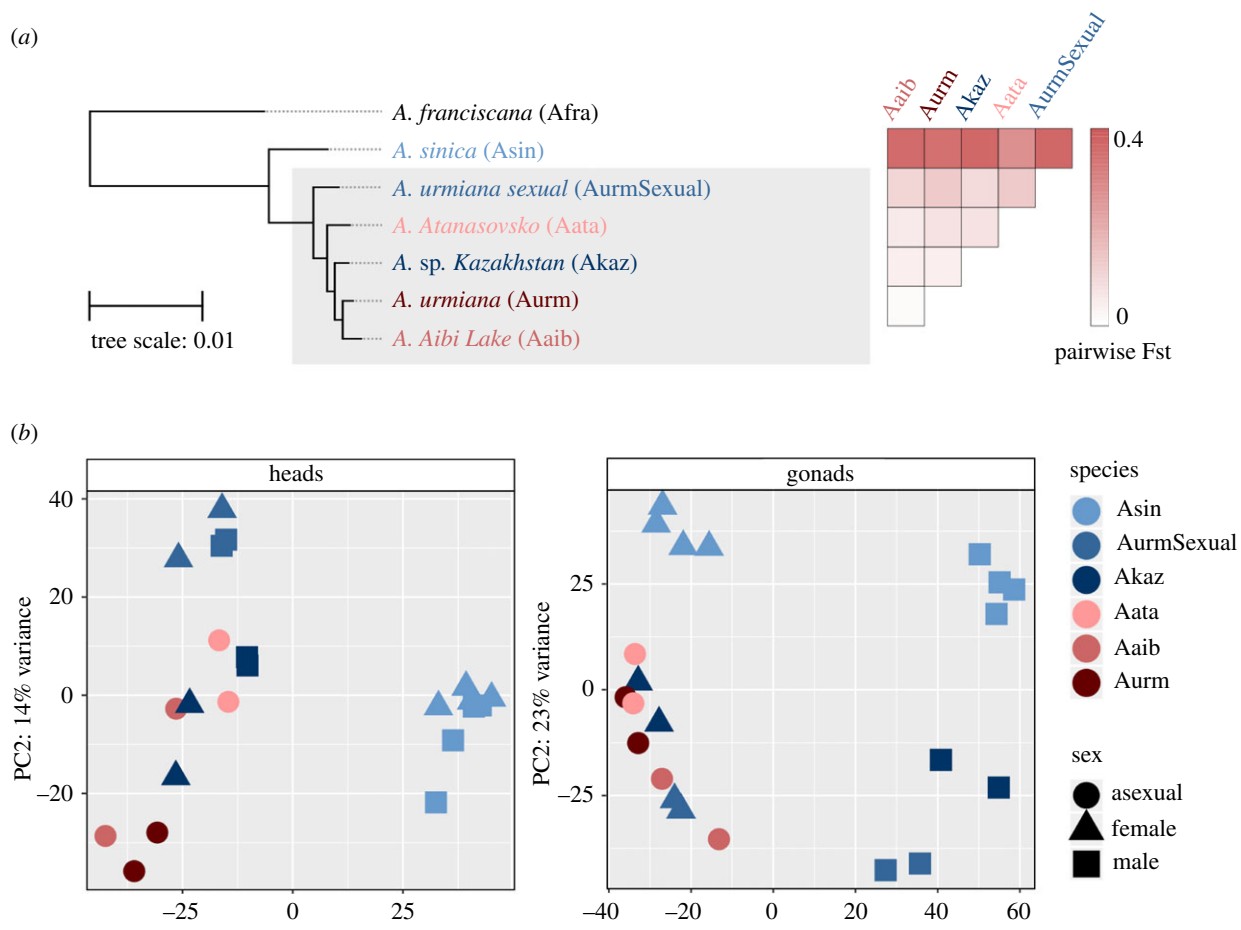

**Figure 1.** Species relationships in *Artemia* based on gene sequence and gene expression data. (*a*) The ML phylogenetic tree of the six Eurasian *Artemia* lineages (asexuals in reds, sexuals in blues) with *A. franciscana* as the outgroup, as well as the mean pairwise Fst estimated from head RNA-seq reads. The grey area indicates very closely related lineages. All branches have 100% bootstrap support. (*b*) The PCA for components 1 and 2 of gene expression in heads and gonads. (Online version in colour.)

(electronic supplementary material, figure S5; RNA-seq for this tissue is not available in *A. urmiana sexual*).

## (b) Small shifts in expression associated with asexuality

We compared *A.* sp. *Kazakhstan* sexual females to asexual females to find differentially expressed genes between the two modes of reproduction using two approaches: first, we pooled asexual females from all three lineages (*A. urmiana*, *A. Aibi Lake* and *A. Atanasovsko*), and compared them jointly to *A.* sp. *Kazakhstan* females (combined analysis); second, we compared each individual asexual lineage to *A.* sp. *Kazakhstan* females and identified genes that showed significant differences in all three comparisons (individual analysis). In the combined analysis, we find 11 up- and 42 downregulated genes in heads and 15 up- and 45 downregulated genes in gonads (electronic supplementary material, table S4). Of these, three are upregulated and nine are downregulated in both tissues. Similarly, although more genes are found to be differentially expressed in the individual comparisons, only 30 (four up, 26 down) and 36 (nine up, 27 down) are found in all three comparisons in heads and gonads, respectively (figure 2; electronic supplementary material, figures S6 and S9). In this case, one gene is upregulated and six are downregulated in both tissues of asexuals, again indicating that only a few genes change their expression consistently in response to asexual reproduction.

Genes that are differentially expressed in at least two of the individual analyses tend to have high fold change differences and very low adjusted *p*-values (figure 2), and the low number of shared genes among all three asexual lineages may be owing to limited statistical power to detect small changes (on the other hand, genes with large expression differences are likely the most biologically relevant).

Although only a few differentially expressed genes in asexuals are shared among the three populations (figure 2), these numbers are higher than expected by chance, indicating that there is a core set of 'asexuality genes', consistent with a single origin of asexuality. Namely, we find four genes upregulated and 26 genes downregulated in heads (versus 0.05 and 0.1 expected, $p = 4.7 \times 10^{-7}$ and $p > 2.2 \times 10^{-16}$, SUPEREXACTTEST). For gonads, nine genes are up- and 27 are downregulated in all three comparisons (versus 0.5 and 0.2 expected, $p < 2.2 \times 10^{-16}$ for both, SUPEREXACTTEST). Forty-eight of these 59 genes were also identified in the combined analysis ('*' electronic supplementary material, tables S5 and S6).

We also tested for differential expression using the more distantly related sexual species *A. urmiana sexual* (electronic supplementary material, figure S7) and *A. sinica* (electronic supplementary material, figure S8) as reference. As expected, we find more differentially expressed genes with increasing phylogenetic distance, but again only a small proportion of differentially expressed genes are shared between the three asexual lineages.

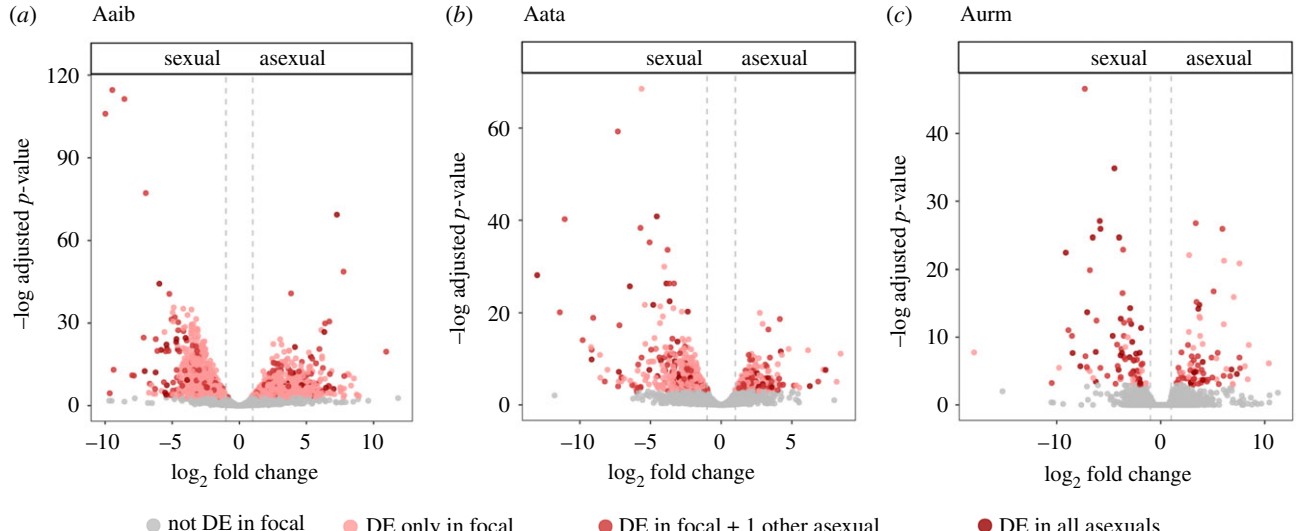

**Figure 2.** Volcano plots for gonad differential gene expression between *A.* sp. *Kazakhstan* females and asexuals for the lineages (*a*) *A. Aibi Lake*, (*b*) *A. Atanasovsko* and (*c*) *A. urmiana*. Significantly differentially expressed (DE) genes are coloured in red and the shade of red shows by how many other asexual lineages this differentially expressed gene is shared. Positive log₂ fold change values mean that a gene is more highly expressed in asexuals than in sexual females and vice versa for negative fold changes. The dashed vertical lines indicate fold change cut-offs of 2. (Online version in colour.)

Genes showing consistent differential expression between sexual and asexual females (combined and individual analyses with *A.* sp. *Kazakhstan* as reference) are not significantly enriched for specific gene ontology (GO) terms (electronic supplementary material, tables S5 and S6). Blast searches of candidates suggest that some of the observed differences may be related to modifications of meiosis in asexual females, similar to Parker *et al.* [23]. For instance, the only gene upregulated in all asexuals in heads and gonads is most similar to a crustacean transitional endoplasmatic reticulum ATPase, a class of genes that, among other functions, is required for chromosome condensation during meiosis [40]. To systematically test for enrichment of meiosis genes, we searched for homologues of annotated meiosis genes from *Drosophila melanogaster* [41] in our *A. sinica* transcriptome, yielding 873 putative homologues. There is no excessive overlap with gonad asexuality genes when asexuals are compared to *A.* sp. *Kazakhstan*. When they are compared to *A. urmiana sexual* or *A. sinica* we find significant enrichment of meiosis genes among the differentially expressed genes in gonads (13 in both comparisons, $p = 4.7 \times 10^{-9}$ and $p = 7.6 \times 10^{-5}$, respectively, Fisher's exact test (FET)) and heads (6 and 23 genes, respectively, $p = 0.006$ and $p > 2.2 \times 10^{-16}$, respectively, FET). These results, therefore, hint at but do not fully confirm a function in meiosis for some of these putative asexuality genes.

## (c) Evolution of sex-biased genes in asexual lineages

Theory predicts that genes with sex-specific functions, and/or those under sex-specific selection are more likely to diverge in expression after shifts to asexuality (see Introduction). We used the outgroup *A. sinica* to define sex-biased genes in three tissues: heads, thorax and gonads. Sex-biased genes were called using DESeq2 (adjusted $p > 0.05$).

We compared females of the three asexual lineages to sexual *A. sinica* females. As a control, we performed the same comparison for sexual *A.* sp. *Kazakhstan* and *A. urmiana sexual* females (except for the thorax in *A. urmiana sexual*). Overall, we observe masculinization of gene expression in the asexuals: genes that are female-biased in *A. sinica* are

downregulated in the asexuals, whereas male-biased genes in *A. sinica* have significantly higher expression in asexual than in *A. sinica* females (figure 3). This is largely consistent across all three tissues (figure 3*a*; electronic supplementary material, figures S10A and S11A), consistent with results in stick insects [23], and the idea that decay of female functions rather than relaxation of sexual antagonism is the dominant force at play. However, this pattern is not specific to asexuals. When we compared female expression of the sexual *A. urmiana sexual* and *A.* sp. *Kazakhstan*, we find the same masculinization of expression (figure 3*a*; all Wilcoxon rank test $p < 0.001$ except for male-biased genes in *A.* sp. *Kazakhstan*). Two hypotheses could account for this: (i) the whole clade may have decreased its investment in female functions, and/or increased its investment in male functions; and (ii) if there is a fast turnover of genes with sex-biased expression, as has been observed in many clades [42,43], female-biased genes with ancestrally high female expression may over time reduce their expression, and vice versa for genes with low male expression, leading to an apparent masculinization of expression when only females are considered.

To test for masculinization of the clade, we analysed the expression in males of the sexual species, which should also show evidence of masculinized expression. In fact, the opposite is observed (figure 3*b*): female-biased genes have increased and male-biased genes have decreased expression in males when comparing *A. urmiana sexual* or *A.* sp. *Kazakhstan* to *A. sinica*. This is significant in gonads, heads and thorax (figure 3*b*; electronic supplementary material, figures S10B and S11B, respectively), with the exception of male-biased genes in gonads and female-biased genes in heads of *A.* sp. *Kazakhstan*, which does not significantly differ from unbiased genes. We also performed a PCA–linear discriminant analysis (LDA) by using gene expression from all males and females from the three sexual species as the training set for a linear model that infers sex and then running the model on the asexuals to estimate their LD1 score (which in this case works as a 'maleness score'). Again, asexuals did not appear masculinized (figure 3*c*); their LD1 distribution overlapped with that of sexual females but was significantly

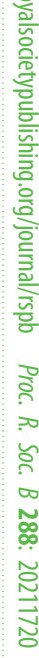

**Figure 3.** Relative gene expression comparisons of Eurasian *Artemia* to the outgroup *A. sinica* in (*a*) female gonads and (*b*) male gonads. Sex-biased genes were defined in *A. sinica* gonads (FBG, female-biased genes; MBG, male-biased genes; UBG, unbiased genes). Female expression is available for all lineages and male expression can only be compared in the sexual lineages. The *y*-axis shows the log expression of the respective species over the expression of *A. sinica*. (*c*) The LDA with linear discriminant 1 on the *y*-axis clearly separating male and female head expression data. The distribution for asexuals overlaps with sexual females. Wilcoxon test: ***$p < 0.001$; n.s., non-significant. (Online version in colour.)

different from that of males. Similar patterns are observed with the head (electronic supplementary material, figure S10C), whereas the thorax had too few samples for the model to be trained correctly (electronic supplementary material, figure S11C).

We also repeated our analysis using only sex-biased genes shared by all three sexual species. This conservative approach ensures that only sex-biased genes that have not undergone turnover are investigated. The gonad results show that, except in *A. Aibi Lake*, higher expression of male-biased genes compared to *A. sinica* can no longer be observed (electronic supplementary material, figure S12), although some reduction of expression of female-biased genes remains. In heads, too few sex-biased genes (11) are conserved for a meaningful analysis. Finally, similar patterns of 'masculinization' are observed when *A.* sp. *Kazakhstan* is used as the reference to call sex-biased genes and as the proxy for ancestral expression (electronic supplementary material, table S1), further arguing against a shift in expression in the whole Kazakhstan clade. Using *A. urmiana sexual* as the reference gives inconsistent results (electronic supplementary material,

table S1), although this seems to be owing to peculiarities of the data or biology for this species, which also yields much fewer sex-biased genes in the gonad. Taken together, these analyses indicate that true masculinization is unlikely to have occurred, whereas fast turnover of sex-biased genes may partly explain our results, and that asexual and sexual species experience similar shifts in gene expression patterns.

## (d) Fast evolution of sex-biased genes in sexual *Artemia*

If the rate of turnover of sex-biased genes is very high, these genes should show fast evolution of gene expression (typically associated with fast sequence evolution [44,45]). To test this, we investigated the expression and sequence divergence of the different classes of sex-biased genes (as defined in *A. sinica* gonads). We find that female-biased genes evolve significantly faster than unbiased genes both in their sequence and gene expression (figure 4*a*; electronic supplementary material, figure S14) in all species (except for sequence divergence in *A. Atanasovsko*, which is non-significant). Male-biased genes have greater expression

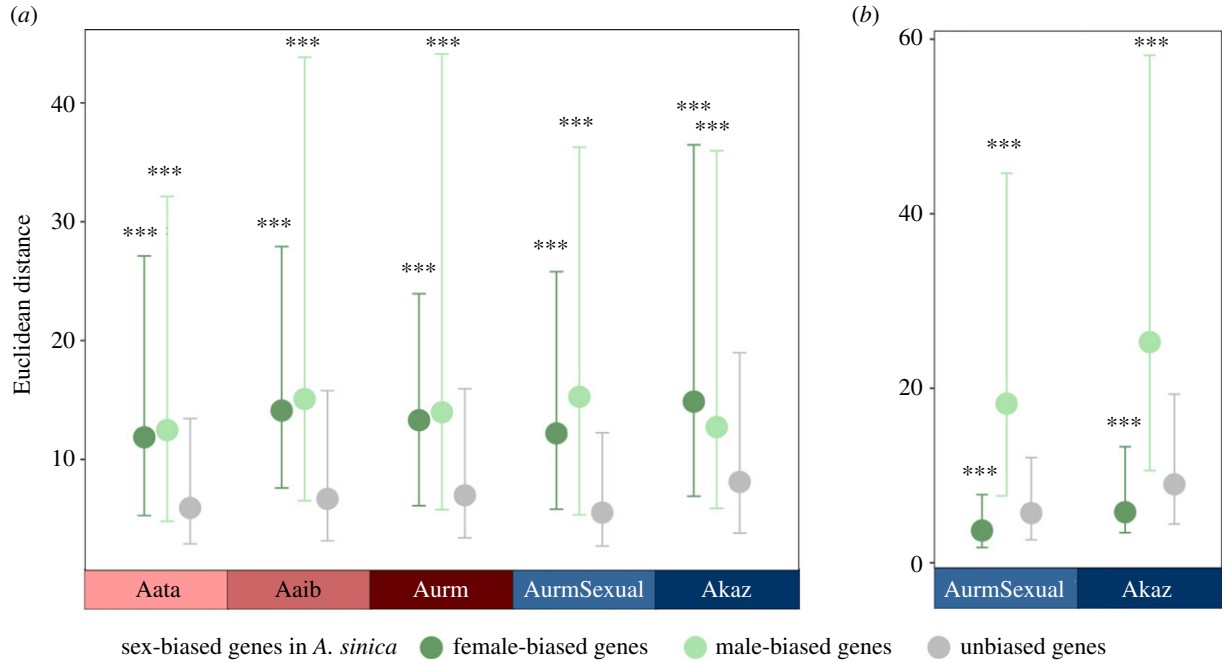

**Figure 4.** Divergence between *A. sinica* and the other five species for sex-biased genes. (*a*) The gene expression divergence in female expression for all species and (*b*) in male expression for the sexual species as Euclidean distances. Sex-biased gene classification is based on *A. sinica* gonads. For Euclidean distance calculation, expression in heads and gonads was used. Wilcoxon test, ***$p < 0.001$. (Online version in colour.)

divergence than both unbiased and female-biased genes, especially when male tissues are included in the calculation (figure 4*b*), but do not show faster evolution of their protein sequences (electronic supplementary material, figure S14). Taken together, this indicates that sex-biased genes, especially female-biased ones, have a fast rate of evolution in the Eurasian *Artemia* species included in this analysis, consistent with turnover contributing to the apparent masculinization of the transcriptome.

## 3. Discussion

### (a) Asexuality does not lead to a major shift in gene expression

It is generally assumed that the transition between sexual and asexual reproduction represents a major evolutionary change [46], potentially requiring a major molecular and cellular reorganization. Previous studies have indeed detected hundreds of genes that are differentially expressed between sexual and asexual lineages [16], or even between sexual and asexual parts of the life cycle of aphids and *Daphnia* [24,47–49]. In the light of this, it is surprising that we find few genes consistently associated with the transition from sexual to asexual reproduction. While it is possible that we do not have enough statistical power to detect genes with minor changes, several biological reasons could be behind this small set of core asexuality genes. First, species with cyclical parthenogenesis typically switch from asexual to sexual reproduction under specific environmental conditions. Genes involved in the response to the change in environments may, therefore, erroneously be detected as differentially expressed between sexual and asexual parts of the life cycle. Second, substantial divergence has often occurred between sexual populations and obligatory parthenogenetic females. Because the lineages studied here are very closely related,

there should be little divergence resulting from relaxed selection in asexuals, and this differentiation may be further eroded by ongoing gene flow among them [37,39]. Third, while some asexual species bypass meiosis altogether (e.g. *Daphnia*, aphids and *Timema* stick insects [50–52]), parthenogenesis in diploid *Artemia* is thought to occur through automixis, i.e. with the occurrence of meiosis [53], leading to more similar patterns of expression between sexual and asexual females. Finally, shifts in gene expression of asexual lineages may occur early in development, as observed in the pea aphid [25]. Future work sampling sexual and asexual *Artemia* throughout their development will provide a full picture of how asexuality is encoded at the molecular level in this clade.

### (b) Turnover of sex-biased genes and masculinization of the asexual transcriptome

We do not observe feminization of gene expression in asexual lineages of Eurasian brine shrimp, as expected if sexual conflict over gene expression is released in the absence of males. On the contrary, asexual gene expression profiles seem masculinized, as observed in stick insects [23]. However, comparing sexual species to each other and including male expression suggests that fast turnover of sex-biased gene expression, and in particular the fast evolution of female-biased genes, is probably responsible for these patterns, rather than true differences between sexual and asexual lineages.

It is possible that true masculinization or feminization does occur in asexual lineages of brine shrimp, but are not detectable in our dataset. These effects could be very small (concerning only a few genes or very small expression changes) or the turnover in sex-biased gene expression could be large enough that it masks signals of feminization/ masculinization. Parker *et al.* [23] used much more divergent sexual–asexual pairs of stick insects, allowing for larger shifts in gene expression to occur, and potentially giving them more power to detect masculinization. On the other

hand, this may have also increased the amount of turnover of genes with sex-specific functions. Asexual stick insects occasionally produce rare males [54], whose expression could potentially be used to confirm the true masculinization of asexual transcriptomes: in this case, rare males should also have increased expression of genes with male functions, and reduced expression of genes with female functions, relative to their male sexual cousins.

## (c) Why do female-biased genes evolve faster in *Artemia*?

Sex-biased genes often diverge faster than genes with similar expressions in both sexes. While male-biased genes are often the fastest-evolving category [42,43], accelerated evolution of female-biased genes has been observed in some species [55–57] and is thought to reflect unusual selective pressures acting on females. For instance, selection on female bird behaviour has been proposed to explain the fast divergence of genes expressed primarily in the brain of the female zebra finch [55]. In mosquitoes, female-biased genes may have evolved rapidly to enable adaptations to blood-feeding [57]. The reproductive life cycle of *Artemia* is also likely to induce strong selection on females, as males use their claspers to grasp females and guard them for days [58]. Mate-guarding may be costly for females, as it can reduce foraging and mating opportunities, potentially leading to widespread sexual conflict and male-female coevolutionary arms-races [59,60]. Direct evidence of this was found in *A. franciscana*, where females have reduced fitness when mated with males derived from cysts with which they have not co-evolved [61].

Finally, brine shrimp have ZW sex determination [36,62–65]. Sex-linked genes often evolve faster than those on autosomes [66,67] and sex-biased genes can be enriched on sex chromosomes [42,43]. Female-biased genes are particularly prone to the 'faster-Z' effect [66,67]. If many of our female-biased genes are Z-specific, this may contribute to their high evolutionary rates. However, the Z-chromosome of the close species *A. franciscana* harbours few sex-biased genes [36]. The faster-Z effect is, therefore, unlikely to explain the high evolutionary rates of female-biased genes, although a full characterization of the sex chromosomes of Eurasian *Artemia* will be necessary to exclude this possibility.

## 4. Methods

A list of all program versions is provided in the electronic supplementary material, table S7. Detailed pipelines are available in the electronic supplementary material, Methods and at: https://git.ist.ac.at/bvicoso/artsexasex.

## (a) Sample collection and sequencing

*Artemia* strains were obtained from the Instituto de Acuicultura de Torre de la Sal (C.S.I.C.) *Artemia* cyst collection in Spain (diploid parthenogenetic: *A. parthenogenetica urmiana*: Urmia Lake (Iran); *A. parthenogenetica Aibi Lake*: Aibi Lake (PR China); *A. parthenogenetica Atanasovsko*: Atanasovsko Lake (Bulgaria). Diploid sexual: *A. sinica*: Tanggu salterns (PR China); *A.* sp. *Kazakhstan*: Kazakhstan unknown locality; *A. urmiana sexual*: Urmia Lake (Iran)). Detailed information is provided in the electronic supplementary material.

Nauplii were hatched in 25°C water with 27 g l$^{-1}$ salinity and then maintained at 30 g l$^{-1}$ salinity under a 14 h : 10 h light : dark cycle. Virgin adults were maintained at 60 g l$^{-1}$ salinity (asexuals produced offspring), and dissected to obtain head and gonad tissue (all species), thorax (all but *A. urmiana sexual*) and whole bodies (*A. sinica, A.* sp. *Kazakhstan, A. Aibi Lake*). Total RNA was extracted from pools of five individuals for each sample using the Bioline Isolate II RNA Mini Kit (cat. no. BIO-52073). At least two biological replicates (using different individuals) were collected per sex and tissue from the first generation to emerge from the cysts (electronic supplementary material, table S2). The RNA-seq samples originally obtained for *A. Aibi Lake* were used for transcriptome assembly but were outliers in the expression analyses. Hence, we sampled and sequenced two new replicates of each tissue (samples 101424–101429 and 101440–101441; electronic supplementary material, table S2), which were used for all downstream expression analyses. Paired-end 125 bp RNA-seq was performed on an Illumina HiSeqV4 at Vienna Biocenter Next Generation Sequencing Core Facility. All RNA-seq libraries were submitted to the NCBI short read archive under bioproject number PRJNA748528.

## (b) Transcriptome assemblies

Transcriptome assemblies for the six Eurasian lineages were performed as described for *A. franciscana* [35]. Briefly, reads were cleaned with Trimmomatic [68] and quality control was performed with FastQC [69]. All available libraries per species were included in the transcriptome assemblies (except for the *A. Aibi Lake* samples obtained later, see the previous section). For each species, we used SOAPdenovo-Trans [70] for multiple K-mers (31–81, step size 10), Trans-ABySS [71] for multiple K-mers (40–84, step size 4) and Trinity [72] for K-mer 25. All SOAPdenovo-Trans assemblies were merged with CD-HIT-EST [73] with a sequence identity threshold of 1.0. Trans-ABySS assemblies were merged with transabyss-merge. Sequences longer than 200 bp in the resulting three assemblies (one per assembler) were combined into one final assembly using the EviGene pipeline [38], yielding a set of transcripts and protein sequences for each species. The number of reads and resulting contigs in all assemblies can be found in the electronic supplementary material, table S3. The quality of our assemblies was assessed with BUSCO [74] (electronic supplementary material, figure S1), using the OrthoDB arthropod reference set [75] with *Daphina pulex* as the reference. All final transcriptome assemblies are available in our git page (https://git.ist.ac.at/bvicoso/artsexasex) and at the IST Austria Data repository (https://doi.org/10.15479/AT:ISTA:9949).

## (c) Orthology and function

Each species was blasted against the *A. sinica* transcriptome and vice versa (BLAST+, [76]). Reciprocal best hits were classified as orthologous genes. This was done on the nucleotide and the amino acid sequences and resulted in 6382 orthologous transcripts and 8389 orthologous proteins found in all seven transcriptomes.

The *A. sinica* transcriptome was functionally annotated using InterProScan [77]. Annotated Pfam domains were used to infer GO terms and topGO [78] was used to assess over- and under-represented GO terms among differentially expressed genes. Significance testing was done with Fisher's exact test using the Benjamini–Hochberg correction for multiple testing at a significance cut-off of *p*-adj > 0.05.

## (d) Phylogenetic analysis

The 8389 1-to-1 orthologous protein sequences from the BBH Blast were used to reconstruct the phylogeny. For ML analysis, each set of orthologous sequences was first aligned with MUSCLE [79] and filtered with Gblocks [80]. Individual alignments for 1-to-1 orthologues were concatenated and phylogenetic reconstruction was

performed using PHYML [81] with 1000 bootstraps. Bayesian inference was also tested using BEAST2 [82]. As ML and Bayesian inference agreed on the tree topology, only the ML results are shown here.

## (e) Single nucleotide polymorphism calling and Fst analysis

Raw RNA-seq reads were mapped to the *A. sinica* filtered transcriptome using BWA, and the resulting sequence alignment/map format (SAM) alignments were sorted using SAMTOOLS [83]. Single nucleotide polymorphisms were called using BCFTOOLS [83] and filtered using VCFTOOLS [84] for a minimum frequency of 0.1, a minimum quality score of 30 and a coverage depth between 10 and 100. We further removed sites with more than two alleles per sample using BCFTOOLS. A customized script was used to calculate Nei's Fst [85] between each pair of lineages from the total number of reads supporting the reference and alternative allele in each species/population.

## (f) Gene expression

*Artemia sinica* transcripts longer than 500 bp were used as a reference (20 888 transcripts). Trimmed RNA-seq reads of all species were mapped to this transcriptome with NEXTGENMAP [86] (see the electronic supplementary material, Methods). Analyses were also performed with the full set of 103 813 transcripts but as results are qualitatively similar, only data from the filtered transcriptome are shown. In addition, reads were mapped to the individual transcriptomes and 1-to-1 orthologues between all species were used to perform differential expression analysis. These results are shown in the electronic supplementary material, figures S13 and S15.

All statistical analyses were performed in R [87]. Principal component and differential gene expression analyses (adjusted $p > 0.05$) were performed using the Bioconductor package DESeq2 [88] separately for each tissue with the Benjamini–Hochberg correction for multiple testing. Expected overlaps between sex-biased genes in sexual species were tested using the SUPEREXACTTEST [89].

Reads per kilobase per million mapped reads (RPKM) values were calculated and normalized across species but separately within each tissue using quantile normalization. Pearson correlations were calculated and samples were hierarchically clustered and visualized using the pheatmap package [90].

For the PCA-LDA analysis, RPKM values were normalized with NORMALYZERDE [91]. We obtained principal components with the R function 'prcomp', and ran a LDA on a subset of these principal components using the R packages MASS [92] and caret [93].

## (g) Divergence

Both non-synonymous substitutions (dN) and synonymous substitutions (dS) values were obtained using the script Script12_KaKs.pl of Picard *et al.* [94]. For each pair of species, reciprocal hits were obtained from the unfiltered transcriptomes using BLAT [95] and aligned with TRANSLATORX [96] with GBLOCKS [80] in codon mode. dN and dS values were estimated using KaKs_calculator [97] with a Jukes–Cantor model of substitution for alignments longer than 500 bp.

Expression divergence for each *A. sinica* transcript was calculated using Euclidean distance [98] from normalized expression values for heads and gonads (averaged biological replicates). Male and female values were treated as separate tissues. Sex-biased genes were compared using the Wilcoxon rank tests.

Data accessibility. All raw RNA-seq data have been uploaded to the NCBI under project PRJNA748528. Processed data files and pipelines are available at: https://git.ist.ac.at/bvicoso/artsexasex.

Authors' contributions. A.K.H.: conceptualization, data curation, formal analysis, methodology, writing—original draft, writing—review and editing; A.M.: methodology, resources; F.H.: resources, writing—original draft, writing—review and editing; B.V.: conceptualization, formal analysis, funding acquisition, project administration, writing—original draft, writing—review and editing. All authors gave final approval for publication and agreed to be held accountable for the work performed therein.

Competing interests. We declare we have no competing interests.

Funding. This work was supported by the European Research Council under the European Union's Horizon 2020 research and innovation program (grant agreement no. 715257).

Acknowledgements. We thank the Vicoso laboratory, Thomas Lenormand and Tanja Schwander for helpful discussions, the group of Gonzalo Gajardo, especially Cristian Gallardo-Escárate and Margarita Parraguez Donoso, for sequencing data and advice, and the IST Scientific Computing Group for their support.

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
