## [Peer Review File · Proceedings of the Royal Society B: Biological Sciences]

Review History

RSPB-2020-1235.R0 (Original submission)

Review form: Reviewer 1

Recommendation

Major revision is needed (please make suggestions in comments)

Scientific importance: Is the manuscript an original and important contribution to its field?

Excellent

General interest: Is the paper of sufficient general interest?

Excellent

Quality of the paper: Is the overall quality of the paper suitable?

Marginal

Is the length of the paper justified?

Yes

Should the paper be seen by a specialist statistical reviewer?

No

Do you have any concerns about statistical analyses in this paper? If so, please specify them explicitly in your report.

Yes

It is a condition of publication that authors make their supporting data, code and materials available - either as supplementary material or hosted in an external repository. Please rate, if applicable, the supporting data on the following criteria.

Is it accessible?

Yes

Is it clear?

Yes

Is it adequate?

Yes

Do you have any ethical concerns with this paper?

No

Comments to the Author

This paper addresses gene expression shifts between sexual and parthenogenetic species of *Artemia* brine shrimp, a fascinating system with several closely related sexual and parthenogenetic species. The study is based on a rich dataset and addresses interesting and timely questions, nicely fitting the scope of ProcB. However, I have concerns regarding how the analyses and comparisons were done, especially because a single sexual species was used as a comparison for all parthenogenetic ones for gene expression analyses. I understand that choosing the best comparisons is not straightforward in the system given the unresolved phylogenetic relationships among the parthenogenetic lineages and the closest sexual relatives, but I do not see how the topology in Figure 1 justifies the chosen single comparisons. I clarify this (and other) points separately for the different manuscript sections.

Part II: Small shifts in expression associated with asexuality

Here the authors want to identify parallel shifts in gene expression between sexual and parthenogenetic females. They compare each of the 3 parthenogenetic species to a single sexual species (Kazakhstan), call DE genes and search for overlap between the different DE gene lists. They find little (but significant) overlap, which they consider “surprising” given that sex-> asex transitions are sometimes hypothesized to “be accompanied by a major reorganisation of gene expression and molecular and cellular organisation”.

I am not particularly surprised that there is little overlap among the 3 pairwise comparisons, given that only 2 replicates are available in each asexual species. This means that there is little power to detect DE genes in each comparison, and even less to detect overlap. I suggest the authors screen for parallel shifts independently of cutoffs for significant differential expression. When addressing the question of which portion of the transcriptome shifts during the sex-asex transition (how “major” is the reorganisation), I would further be interested in the results of individual comparisons, not the overlap among different shifts.

The 3 pairwise comparisons are of course not mutually independent since they all involve the same sexual species (the authors address this point by pooling the asexuals). But I do not understand why the authors do not use different sexual species for comparison, since they are available. If every asexual is compared to all three sexual species for identifying DE genes, the authors could give a range of genes (min-max) with parallel changes, depending on the set of sexual species used for comparison (or at least the two closely sexual species could be used instead of just one).

Part III: Evolution of sex-biased genes in asexual lineages

In this part of the ms, the authors study the fate of sex-biased genes in sex-asex transitions, again, via pairwise comparisons with only 1 sexual species (which is also used to categorize genes into M-biased, F-biased and unbiased). Surprisingly, they do not use the same sexual species as in the previous part, but a more distantly related, sexual “outgroup” (*A. sinica*). Again, why not use the two closely related sexual species (Kazakhstan and urmiana) to study the fate of genes that are sex-biased in either one of these species in the three asexuals? This would allow for a discussion of patterns independently of shifts between the outgroup and the clade comprising sexual and asexual species highlighted in grey in fig 1.

Part IV. Fast evolution of sex-biased genes in sexual *Artemia*

In this section I am not sure that the finding of faster divergence of expression of female-biased genes is a biological results rather than a technical artifact. Although we know from other studies that there is indeed a (weak) correlation between the rate of sequence and expression evolution, the present finding might be a consequence of the methods used.

The authors quantify gene expression by mapping reads from all species to the sexual outgroup (*A. sinica*). This means that faster evolving genes (such as F-biased genes in the present study) will have a lower mapping probability than rel. conserved genes. Reduced mapping probability would appear as reduced expression of F-biased genes in asexuals (and increased expression divergence) even if it was not the case.

At least as supplemental information, I would like to see the same analyses done with mapping reads from each species to its own transcriptome, and inferring orthologs among species (not just for this section but also for the other analyses in fact).

Part I: Phylogeny & evolution of asexuality

This part has some strange methodology. First, why would the authors solely use “head samples” for calling SNPs?. Since each sample is a pool of 5 individuals (presumably different ones?), I would pool all samples per species to call SNPs, as this would give the most accurate estimate of allele frequencies in a given lineage.

After calling SNPs in each sample, the authors run the program structure to find genetic clusters. However, structure treats samples as individuals (with 0/0 1/1 or 1/1 genotypes), not taking into account allele frequencies at each locus. I would completely change this section and use a pool-seq analytical approach based on allele frequencies rather than genotypes.

Other comments.

-There is insufficient and sometimes confusing information about the species and biological material used. For example, how genetically diverse are the strains given they were obtained from some type of “stock center”? I was surprised to see so much assignment variation among different pools of individuals from a single strain (Figure 1 B, Suppl. Fig. 2) – especially for the asexual strains. In some cases (notably Aata) I was also wondering if the authors verified that these were in fact diploid and not polyploid? Why did the authors not use the transcriptome data from “thorax” indicated in the methods?

-The authors state that homologs of genes differently expressed between sexual and asexual females in both heads and gonads “are all involved in meiosis and/or cell division [2], suggesting that they may play a role in the modified meiosis required for asexual reproduction in brine shrimp”. I think this is a bit of an over-interpretation since there are no meiotic divisions in the head.

Review form: Reviewer 2

Recommendation

Major revision is needed (please make suggestions in comments)

Scientific importance: Is the manuscript an original and important contribution to its field?

Marginal

General interest: Is the paper of sufficient general interest?

Acceptable

Quality of the paper: Is the overall quality of the paper suitable?

Marginal

Is the length of the paper justified?

Yes

Should the paper be seen by a specialist statistical reviewer?

No

Do you have any concerns about statistical analyses in this paper? If so, please specify them explicitly in your report.

No

It is a condition of publication that authors make their supporting data, code and materials available - either as supplementary material or hosted in an external repository. Please rate, if applicable, the supporting data on the following criteria.

Is it accessible?

Yes

Is it clear?

Yes

Is it adequate?

Yes

Do you have any ethical concerns with this paper?

No

Comments to the Author

This manuscript looks at patterns of gene expression in male and female brine shrimp of sexual species as well as the gene expression in closely related asexual lineages. One aim is to identify genes with expression patterns that change in a consistent way during the transition to asexuality, which could be informative about how this transition occurs functionally, and the authors find some genes that seem to show consistent changes. Another interesting aim which the Introduction spends quite a bit of time on is to test ideas about how relaxation of sex-specific selective pressures alters gene expression patterns during the transition to asexuality, but the results here are a bit difficult to interpret, although that may be able to be addressed. My comments below.

- For the PC analysis, do other PCs show anything interesting? For example, in Fig. 1, do heads separate by sex on one of the later PCs? There is plenty of variance still to explain after 1/2. If there is an axis that seems to separate male and female samples in the sexual lineages, and I think it would be weird if there wasn't, you may find an interesting/informative pattern in how the asexuals show up on that PC.

- On line 174, what are the chance expectations? And report the stats here for the comparison between observed and expected. This is actually a fairly important statement and the support needs to be shared.

- l. 196-200 – At least as written currently, I think this is a pretty tenuous connection. Just from a quick look, the specific gene mentioned here seems to have diverse functions outside of meiosis. It may be that there is a stronger link to be made between meiosis and this gene (and for the 21 out of 68 mentioned previously), but as I read the text now that link does not seem convincing.

- Most of my comments are related to Part III, which to me is the most important/interesting section in light of the setup of the paper. After thinking about this section for a while, particularly the concluding sentences, isn't the main conclusion here that all the species along the branch you're interested in using to test these ideas about sexuality (2 species) versus asexuality (3 species) are in fact showing less sex-biased gene expression than the outgroup, and are fairly similar in masculinity/femininity of gene expression? If that is the case it's not a particularly satisfying test of the idea, because as the authors write here it could just be a feature of the Eurasian artemia. Then we don't know whether or not the lack of masculinization or feminization is due to a lack of the putative selective forces that would cause that or instead (for example) a lack of time for masculinization or feminization to take place...

Related to my earlier comment about the PC visualization, it seems to me like a better way to test the idea in the Introduction – do asexuals look feminized or masculinized – would be to approach this in a multivariate way. For example, you could do a PCA on a sexual lineage, or both (e.g. urm, or both urm and kaz), males and females, which will give you a clear separation of males and females on PC1. Then plot the asexuals on the same PC to see if they fall out as intermediate, more male-like, more female-like, 'uber' female, 'uber' male, etc. This could be done for each tissue and is a direct test of whether the asexuals are feminized or masculinized or neither. It allows you to pool information about the overall profile of maleness and femaleness, rather than looking gene by gene, and also allows you to be more quantitative about it, because it assigns each of the samples a numeric masculinity or femininity. Maybe it would be better to use discriminant analysis, actually, to specifically define male vs. female in urmiana or urmiana+kazakhstan, and then plot your other species on the discriminant axes. Right now as I understand it we are limited to interpretation of Figure 3, which says that they're all kinda sorta going the same way relative to *A. sinica*.

Also, given the discussion of the observed rapid turnover of sex-biased genes in other groups, why use *A. sinica* to define sex-biased genes? According to the phylogeny, it seems like you should be able to use *A. urmiana* to define the genes, and the result would be a much more consistent pattern of a male- or female-biased gene where the other species you're interested in much more consistently show the same bias.

- I don't see any mention of this, but it is probably worth discussing whether allometry could impact the results – are the tissues being profiled wildly divergent in sizes, or the sizes of subcomponents?

- L. 258-289 says, "If the rate of turnover of sex-biased genes in the Eurasian Artemia species is very high, these genes should show signatures of fast evolution." Is that true? Wouldn't it be more likely actually that the turnover is caused by regulatory regions outside of the genes, and that is where you would see the sequence-level changes? Also, don't sex-biased genes in general, across taxonomic groups, show signatures of fast evolution? In that case you would then expect to see this signal that confirms a widespread pattern regardless of the specific rates of turnover occurring here.

- Some smaller things:

- l. 140 – "We head..." should be "We profiled head..."

The figure 1 legend needs to be elaborated for clarity and written less informally. Also principle should be principal.

l. 178-179 – I’m a little skeptical of $1e-53$ and $1e-18$ p values in this test... How do we interpret $1e-53$, really? It seems likely that the statistical test is inappropriate.

Same problem with figure 2 legend, too informal. E.g. “A) shows up-regulated and B) down-regulated genes in asexuals heads.”

Figure 3 also needs work. For example, the legend begins with “Gene expression in...” but this figure shows relative gene expression. It would also be nice to have something clarifying A is females and B is males without having to read the legend (for example, male and female symbols or the words male and female above the panels). And the y axis is hard to parse, actually, maybe there is some way to make this more readable while also making it clear that every single plot here is a comparison to *A. sinica* counterparts?

Review form: Reviewer 3

Recommendation

Major revision is needed (please make suggestions in comments)

Scientific importance: Is the manuscript an original and important contribution to its field?

Good

General interest: Is the paper of sufficient general interest?

Excellent

Quality of the paper: Is the overall quality of the paper suitable?

Acceptable

Is the length of the paper justified?

Yes

Should the paper be seen by a specialist statistical reviewer?

No

Do you have any concerns about statistical analyses in this paper? If so, please specify them explicitly in your report.

Yes

It is a condition of publication that authors make their supporting data, code and materials available - either as supplementary material or hosted in an external repository. Please rate, if applicable, the supporting data on the following criteria.

Is it accessible?

Yes

Is it clear?

Yes

Is it adequate?

Yes

Do you have any ethical concerns with this paper?

No

Comments to the Author

In this article, Huylmans and colleagues measure gene expression in 3 sexual and 3 asexual lineages (or species) in *Artemia* brine shrimps, to study how gene expression is affected by transitions to asexuality. They identify genes systematically less or more expressed in asexuals than in sexuals, and show that meiosis genes are overrepresented in this list. They also test whether the release of sexual conflict between males and females (sexual conflicts do not occur anymore in asexual lineages) affects the evolution of gene expression.

Main comments:

Meiosis genes: The authors show that meiosis genes are common in the set of genes differentially expressed between sexual and asexual as 21 out of 68 genes have a meiosis-related function. But they do not explain how they determine that a gene has a meiosis-related function, and the 21 genes are not identified in the supplementary tables. The authors should also demonstrate statistically that 21 out of 68 is more than expected (thus measure the number of genes with meiosis-related functions in the list of gene that are not DE between sexuals and asexuals).

Masculinization of expression: I wonder if the fact that female-biased genes (identified in a sexual species A) tend to be systematically less female-biased in the other species is not an expected consequence of the way the analyses are done. Let's suppose that with RNAseq we just get an estimate of the true level of expression for a gene (so this estimate is of course imperfect). Hence, those genes identified as female-biased in species A are those genes that were the most female-biased in the noisy RNAseq data (female-bias is due to true female-biased expression effect but also to some stochastic effects that may have increase the bias). So, by construction, when looking at the expression of female-biased genes defined in species A in another species, in most cases these genes will be less female-biased. I have no idea if the effect is tiny or not, but I think it adds to the other mechanisms hypothesized by the authors.

Origin of samples: More information on the geographical origin and areas of distribution of the different species should be provided.

Phylogeny: No branch support is provided for the phylogeny (Figure 1A). Such information is important especially as the authors compare the clustering of species to a previous study (lines 121-122).

RNAseq samples: It is not clear if a single clone was used for asexual species or if different clones were pooled. Similarly, it is also not clear if RNAseq replicates correspond to the same pool of 5 individuals (sequenced twice) or to 2 pools of 5 different individuals.

Admixture analyses of the RNAseq samples: This method is not adapted to the data for different reasons. The admixture analysis has been developed in a population genetic framework. This analysis clusters samples (individuals) into populations by minimizing Hardy-Weinberg disequilibrium within the inferred populations (hence per population allelic frequencies are estimated) and assuming linkage equilibrium. Thus many samples (individuals) originating from different populations are required. This is clearly not the case with the data analyzed here (a few samples per species). Furthermore, asexual lineages usually do not satisfy HW assumptions and linkage equilibrium. A possibility to look at the relationship between samples would be to construct a tree based on population genetic distances between samples.

Furthermore, since there is no precise description of the content of the RNAseq samples, I have additional questions. I suppose that each sample (for sexual species) is a mix of 5 genetically different individuals, but then each sample is analyzed as a diploid sample for SNPs calling. This is weird. Are the two replicates for a tissue sequencing replicates or do the individuals used differs? (this will affect the similarity of the replicates).

Lines 50-51: "..., while other studies found much more subtle differences in expression between sexual and asexual lineages [25-27]." None of the 3 cited articles are about gene expression.

Minor comments:

The aim of the study in the introduction is vague (line 101: "... allowing us to test some of the questions on the evolution of asexuality.")

Two A. parthenogenetica samples were removed. Could you mark them on the suppl fig 3 and 4 (I see more than two A. parth samples that do not cluster so well).

Line 258: Why expect fast evolution if rate of turnover a sex-biased gene is high?

Decision letter (RSPB-2020-1235.R0)

07-Jul-2020

Dear Dr Huylmans:

I am writing to inform you that your manuscript RSPB-2020-1235 entitled "Transitions to asexuality and evolution of gene expression in Artemia brine shrimp" has, in its current form, been rejected for publication in Proceedings B.

This action has been taken on the advice of referees, who have recommended that substantial revisions are necessary. With this in mind we would be happy to consider a resubmission, provided the comments of the referees are fully addressed. However please note that this is not a provisional acceptance.

Sincerely,
 Professor Gary Carvalho
 mailto: proceedingsb@royalsociety.org

Associate Editor

Comments to Author:

All three reviewers were generally enthusiastic, although there were several important concerns raised about the phylogenetic and other methods.

Reviewer 1 has several questions, most importantly with the strategy of using a single sexual species as a comparison for all parthenogenetic ones for gene expression analyses. Reviewer 2 makes several helpful suggestions for further analyses that will help the authors to make more concrete conclusions. Reviewer 3 raises a very interesting idea that the turnover of sex biased genes and the masculinization observed in asexual lineages might in fact be a product of regression toward the mean rather than a real biological process, and it would be very helpful if the authors can confirm or refute this. We had a similar issue with a previous analysis (Pointer et al. PLOS Genetics 2013) that might be helpful.

I had a few additional questions not raised by the reviewers:

How certain is that the asexual lineages are fully and completely asexual? The Structure plot in Fig 1B calls one of the Atanasovsko populations into question. How was asexuality determined in these lineages?

Also, with regard to Fig 1B - I assume some species have multiple columns because they were sampled from multiple populations? If so, it might help reduce reader confusion to mention this in the figure legend.

Reviewer(s)' Comments to Author:

Referee: 1

Comments to the Author(s)

This paper addresses gene expression shifts between sexual and parthenogenetic species of *Artemia* brine shrimp, a fascinating system with several closely related sexual and parthenogenetic species. The study is based on a rich dataset and addresses interesting and timely questions, nicely fitting the scope of ProcB. However, I have concerns regarding how the analyses and comparisons were done, especially because a single sexual species was used as a comparison for all parthenogenetic ones for gene expression analyses. I understand that choosing the best comparisons is not straightforward in the system given the unresolved phylogenetic relationships among the parthenogenetic lineages and the closest sexual relatives, but I do not see how the topology in Figure 1 justifies the chosen single comparisons. I clarify this (and other) points separately for the different manuscript sections.

Part II: Small shifts in expression associated with asexuality

Here the authors want to identify parallel shifts in gene expression between sexual and parthenogenetic females. They compare each of the 3 parthenogenetic species to a single sexual species (Kazakhstan), call DE genes and search for overlap between the different DE gene lists. They find little (but significant) overlap, which they consider "surprising" given that sex-> asex transitions are sometimes hypothesized to "be accompanied by a major reorganisation of gene expression and molecular and cellular organisation".

I am not particularly surprised that there is little overlap among the 3 pairwise comparisons, given that only 2 replicates are available in each asexual species. This means that there is little power to detect DE genes in each comparison, and even less to detect overlap. I suggest the authors screen for parallel shifts independently of cutoffs for significant differential expression. When addressing the question of which portion of the transcriptome shifts during the sex-asex transition (how "major" is the reorganisation), I would further be interested in the results of individual comparisons, not the overlap among different shifts.

The 3 pairwise comparisons are of course not mutually independent since they all involve the same sexual species (the authors address this point by pooling the asexuals). But I do not understand why the authors do not use different sexual species for comparison, since they are

available. If every asexual is compared to all three sexual species for identifying DE genes, the authors could give a range of genes (min-max) with parallel changes, depending on the set of sexual species used for comparison (or at least the two closely sexual species could be used instead of just one).

Part III: Evolution of sex-biased genes in asexual lineages

In this part of the ms, the authors study the fate of sex-biased genes in sex-asex transitions, again, via pairwise comparisons with only 1 sexual species (which is also used to categorize genes into M-biased, F-biased and unbiased). Surprisingly, they do not use the same sexual species as in the previous part, but a more distantly related, sexual “outgroup” (*A. sinica*). Again, why not use the two closely related sexual species (Kazakhstan and urmiana) to study the fate of genes that are sex-biased in either one of these species in the three asexuals? This would allow for a discussion of patterns independently of shifts between the outgroup and the clade comprising sexual and asexual species highlighted in grey in fig 1.

Part IV. Fast evolution of sex-biased genes in sexual *Artemia*

In this section I am not sure that the finding of faster divergence of expression of female-biased genes is a biological results rather than a technical artifact. Although we know from other studies that there is indeed a (weak) correlation between the rate of sequence and expression evolution, the present finding might be a consequence of the methods used.

The authors quantify gene expression by mapping reads from all species to the sexual outgroup (*A. sinica*). This means that faster evolving genes (such as F-biased genes in the present study) will have a lower mapping probability than rel. conserved genes. Reduced mapping probability would appear as reduced expression of F-biased genes in asexuals (and increased expression divergence) even if it was not the case.

At least as supplemental information, I would like to see the same analyses done with mapping reads from each species to its own transcriptome, and inferring orthologs among species (not just for this section but also for the other analyses in fact).

Part I: Phylogeny & evolution of asexuality

This part has some strange methodology. First, why would the authors solely use “head samples” for calling SNPs?. Since each sample is a pool of 5 individuals (presumably different ones?), I would pool all samples per species to call SNPs, as this would give the most accurate estimate of allele frequencies in a given lineage.

After calling SNPs in each sample, the authors run the program structure to find genetic clusters. However, structure treats samples as individuals (with 0/0 1/1 or 1/1 genotypes), not taking into account allele frequencies at each locus. I would completely change this section and use a pool-seq analytical approach based on allele frequencies rather than genotypes.

Other comments.

-There is insufficient and sometimes confusing information about the species and biological material used. For example, how genetically diverse are the strains given they were obtained from some type of “stock center”? I was surprised to see so much assignment variation among different pools of individuals from a single strain (Figure 1 B, Suppl. Fig. 2) – especially for the asexual strains. In some cases (notably Aata) I was also wondering if the authors verified that these were in fact diploid and not polyploid? Why did the authors not use the transcriptome data from “thorax” indicated in the methods?

-The authors state that homologs of genes differently expressed between sexual and asexual females in both heads and gonads “are all involved in meiosis and/or cell division [2], suggesting

that they may play a role in the modified meiosis required for asexual reproduction in brine shrimp". I think this is a bit of an over-interpretation since there are no meiotic divisions in the head.

Referee: 2

Comments to the Author(s)

This manuscript looks at patterns of gene expression in male and female brine shrimp of sexual species as well as the gene expression in closely related asexual lineages. One aim is to identify genes with expression patterns that change in a consistent way during the transition to asexuality, which could be informative about how this transition occurs functionally, and the authors find some genes that seem to show consistent changes. Another interesting aim which the Introduction spends quite a bit of time on is to test ideas about how relaxation of sex-specific selective pressures alters gene expression patterns during the transition to asexuality, but the results here are a bit difficult to interpret, although that may be able to be addressed. My comments below.

- For the PC analysis, do other PCs show anything interesting? For example, in Fig. 1, do heads separate by sex on one of the later PCs? There is plenty of variance still to explain after 1/2. If there is an axis that seems to separate male and female samples in the sexual lineages, and I think it would be weird if there wasn't, you may find an interesting/informative pattern in how the asexuals show up on that PC.
- On line 174, what are the chance expectations? And report the stats here for the comparison between observed and expected. This is actually a fairly important statement and the support needs to be shared.
- l. 196-200 – At least as written currently, I think this is a pretty tenuous connection. Just from a quick look, the specific gene mentioned here seems to have diverse functions outside of meiosis. It may be that there is a stronger link to be made between meiosis and this gene (and for the 21 out of 68 mentioned previously), but as I read the text now that link does not seem convincing.
- Most of my comments are related to Part III, which to me is the most important/interesting section in light of the setup of the paper. After thinking about this section for a while, particularly the concluding sentences, isn't the main conclusion here that all the species along the branch you're interested in using to test these ideas about sexuality (2 species) versus asexuality (3 species) are in fact showing less sex-biased gene expression than the outgroup, and are fairly similar in masculinity/femininity of gene expression? If that is the case it's not a particularly satisfying test of the idea, because as the authors write here it could just be a feature of the Eurasian artemia. Then we don't know whether or not the lack of masculinization or feminization is due to a lack of the putative selective forces that would cause that or instead (for example) a lack of time for masculinization or feminization to take place...

Related to my earlier comment about the PC visualization, it seems to me like a better way to test the idea in the Introduction – do asexuals look feminized or masculinized – would be to approach this in a multivariate way. For example, you could do a PCA on a sexual lineage, or both (e.g. urm, or both urm and kaz), males and females, which will give you a clear separation of males and females on PC1. Then plot the asexuals on the same PC to see if they fall out as intermediate, more male-like, more female-like, 'uber' female, 'uber' male, etc. This could be done for each tissue and is a direct test of whether the asexuals are feminized or masculinized or neither. It allows you to pool information about the overall profile of maleness and femaleness, rather than looking gene by gene, and also allows you to be more quantitative about it, because it assigns each of the samples a numeric masculinity or femininity. Maybe it would be better to use discriminant analysis, actually, to specifically define male vs. female in urmiana or urmiana+kazakhstan, and then plot your other species on the discriminant axes. Right now as I understand it we are limited to interpretation of Figure 3, which says that they're all kinda sorta going the same way relative to *A. sinica*.

Also, given the discussion of the observed rapid turnover of sex-biased genes in other groups, why use *A. sinica* to define sex-biased genes? According to the phylogeny, it seems like you

should be able to use *A. urmiana* to define the genes, and the result would be a much more consistent pattern of a male- or female-biased gene where the other species you're interested in much more consistently show the same bias.

- I don't see any mention of this, but it is probably worth discussing whether allometry could impact the results—are the tissues being profiled wildly divergent in sizes, or the sizes of subcomponents?
- L. 258-289 says, "If the rate of turnover of sex-biased genes in the Eurasian *Artemia* species is very high, these genes should show signatures of fast evolution." Is that true? Wouldn't it be more likely actually that the turnover is caused by regulatory regions outside of the genes, and that is where you would see the sequence-level changes? Also, don't sex-biased genes in general, across taxonomic groups, show signatures of fast evolution? In that case you would then expect to see this signal that confirms a widespread pattern regardless of the specific rates of turnover occurring here.

• Some smaller things:

l. 140 – "We head..." should be "We profiled head..."

The figure 1 legend needs to be elaborated for clarity and written less informally. Also principle should be principal.

l. 178-179 – I'm a little skeptical of $1e-53$ and $1e-18$ p values in this test... How do we interpret $1e-53$, really? It seems likely that the statistical test is inappropriate.

Same problem with figure 2 legend, too informal. E.g. "A) shows up-regulated and B) down-regulated genes in asexuals heads."

Figure 3 also needs work. For example, the legend begins with "Gene expression in..." but this figure shows relative gene expression. It would also be nice to have something clarifying A is females and B is males without having to read the legend (for example, male and female symbols or the words male and female above the panels). And the y axis is hard to parse, actually, maybe there is some way to make this more readable while also making it clear that every single plot here is a comparison to *A. sinica* counterparts?

Referee: 3

Comments to the Author(s)

In this article, Huylmans and colleagues measure gene expression in 3 sexual and 3 asexual lineages (or species) in *Artemia* brine shrimps, to study how gene expression is affected by transitions to asexuality. They identify genes systematically less or more expressed in asexuals than in sexuals, and show that meiosis genes are overrepresented in this list. They also test whether the release of sexual conflict between males and females (sexual conflicts do not occur anymore in asexual lineages) affects the evolution of gene expression.

Main comments:

Meiosis genes: The authors show that meiosis genes are common in the set of genes differentially expressed between sexual and asexual as 21 out of 68 genes have a meiosis-related function. But they do not explain how they determine that a gene has a meiosis-related function, and the 21 genes are not identified in the supplementary tables. The authors should also demonstrate statistically that 21 out of 68 is more than expected (thus measure the number of genes with meiosis-related functions in the list of gene that are not DE between sexuals and asexuals).

Masculinization of expression: I wonder if the fact that female-biased genes (identified in a sexual species A) tend to be systematically less female-biased in the other species is not an expected consequence of the way the analyses are done. Let's suppose that with RNAseq we just get an estimate of the true level of expression for a gene (so this estimate is of course imperfect). Hence, those genes identified as female-biased in species A are those genes that were the most female-biased in the noisy RNAseq data (female-bias is due to true female-biased expression effect but also to some stochastic effects that may have increase the bias). So, by construction, when looking

at the expression of female-biased genes defined in species A in another species, in most cases these genes will be less female-biased. I have no idea if the effect is tiny or not, but I think it adds to the other mechanisms hypothesized by the authors.

Origin of samples: More information on the geographical origin and areas of distribution of the different species should be provided.

Phylogeny: No branch support is provided for the phylogeny (Figure 1A). Such information is important especially as the authors compare the clustering of species to a previous study (lines 121-122).

RNAseq samples: It is not clear if a single clone was used for asexual species or if different clones were pooled. Similarly, it is also not clear if RNAseq replicates correspond to the same pool of 5 individuals (sequenced twice) or to 2 pools of 5 different individuals.

Admixture analyses of the RNAseq samples: This method is not adapted to the data for different reasons. The admixture analysis has been developed in a population genetic framework. This analysis clusters samples (individuals) into populations by minimizing Hardy-Weinberg disequilibrium within the inferred populations (hence per population allelic frequencies are estimated) and assuming linkage equilibrium. Thus many samples (individuals) originating from different populations are required. This is clearly not the case with the data analyzed here (a few samples per species). Furthermore, asexual lineages usually do not satisfy HW assumptions and linkage equilibrium. A possibility to look at the relationship between samples would be to construct a tree based on population genetic distances between samples.

Furthermore, since there is no precise description of the content of the RNAseq samples, I have additional questions. I suppose that each sample (for sexual species) is a mix of 5 genetically different individuals, but then each sample is analyzed as a diploid sample for SNPs calling. This is weird. Are the two replicates for a tissue sequencing replicates or do the individuals used differs? (this will affect the similarity of the replicates).

Lines 50-51: "... , while other studies found much more subtle differences in expression between sexual and asexual lineages [25-27]." None of the 3 cited articles are about gene expression.

Minor comments:

The aim of the study in the introduction is vague (line 101: "... allowing us to test some of the questions on the evolution of asexuality.")

Two *A. parthenogenetica* samples were removed. Could you mark them on the suppl fig 3 and 4 (I see more than two *A. parth* samples that do not cluster so well).

Line 258: Why expect fast evolution if rate of turnover a sex-biased gene is high?

Author's Response to Decision Letter for (RSPB-2020-1235.R0)

See Appendix A.

RSPB-2021-1720.R0

Review form: Reviewer 1

Recommendation

Accept with minor revision (please list in comments)

Scientific importance: Is the manuscript an original and important contribution to its field?

Excellent

General interest: Is the paper of sufficient general interest?

Excellent

Quality of the paper: Is the overall quality of the paper suitable?

Excellent

Is the length of the paper justified?

Yes

Should the paper be seen by a specialist statistical reviewer?

No

Do you have any concerns about statistical analyses in this paper? If so, please specify them explicitly in your report.

No

It is a condition of publication that authors make their supporting data, code and materials available - either as supplementary material or hosted in an external repository. Please rate, if applicable, the supporting data on the following criteria.

Is it accessible?

Yes

Is it clear?

Yes

Is it adequate?

Yes

Do you have any ethical concerns with this paper?

No

Comments to the Author

I think the authors have promptly addressed all comments raised on the previous version. I only have two more minor comments.

1) In L148-150, the authors argue that “genes with large expression differences are likely the most biologically relevant, and the fact that so few consistently change in asexuals suggests that the shift in reproductive mode may not entail a major reorganization of the cell machinery”. In arthropods, the vast majority of genes expressed in female gonads are linked to oocyte differentiation (vitellogenesis etc). There is a very small part in the gonads (the germarium) where meiosis actually happens, and meiosis-related genes have generally very low expression in the germarium of sexual females (and sometimes meiosis actually happens in juvenile stages). This means that there could still be a major reorganization of the cell machinery linked to parthenogenetic reproduction, but such a reorganization could not be detected in the dataset

because it would manifest as minor gene expression differences. I suggest omitting or reformulating this statement.

2) Could the authors clarify in the methods whether sexual females were virgin or mated (this has major effects on females in some species, eg *Drosophila melanogaster*), and whether sexual and asexual females were collected while reproductively active? (In many arthropod species, there is a time lag between the moult leading to sexual maturity and the onset of reproduction). For example, little gene expression difference between sexual and asexual females would perhaps be the expected pattern if females were not yet reproductively active.

Decision letter (RSPB-2021-1720.R0)

20-Aug-2021

Dear Dr Huylmans:

Your manuscript has now been peer reviewed and the reviews have been assessed by an Associate Editor. The reviewers' comments (not including confidential comments to the Editor) and the comments from the Associate Editor are included at the end of this email for your reference. As you will see, the reviewers and the Editors have raised some concerns with your manuscript and we would like to invite you to revise your manuscript to address them.

Research ethics:

Use of animals and field studies:

If your study uses animals please include details in the methods section of any approval and licences given to carry out the study and include full details of how animal welfare standards

were ensured. Field studies should be conducted in accordance with local legislation; please include details of the appropriate permission and licences that you obtained to carry out the field work.

It is a condition of publication that you make available the data and research materials supporting the results in the article (<https://royalsociety.org/journals/authors/author-guidelines/#data>). Datasets should be deposited in an appropriate publicly available repository and details of the associated accession number, link or DOI to the datasets must be included in the Data Accessibility section of the article (<https://royalsociety.org/journals/ethics-policies/data-sharing-mining/>). Reference(s) to datasets should also be included in the reference list of the article with DOIs (where available).

If you wish to submit your data to Dryad (<http://datadryad.org/>) and have not already done so you can submit your data via this link [http://datadryad.org/submit?journalID=RSPB&manu=\(Document not available\)](http://datadryad.org/submit?journalID=RSPB&manu=(Document%20not%20available)), which will take you to your unique entry in the Dryad repository.

Please submit a copy of your revised paper within three weeks. If we do not hear from you within this time your manuscript will be rejected. If you are unable to meet this deadline please let us know as soon as possible, as we may be able to grant a short extension.

Best wishes,
Professor Gary Carvalho
<mailto:proceedingsb@royalsociety.org>

Associate Editor

Comments to Author:

One of the original reviewers and myself have read through the revised manuscript, and are very happy with the new version. I would like to thank the authors for their careful responses to the suggestions in the previous round of review, and their thorough revision. The reviewer has two

minor comments that should be very easy to deal with, and I do not expect to send a revised version back to review.

Reviewer(s)' Comments to Author:

Referee: 1

Comments to the Author(s).

I think the authors have promptly addressed all comments raised on the previous version. I only have two more minor comments.

1) In L148-150, the authors argue that “genes with large expression differences are likely the most biologically relevant, and the fact that so few consistently change in asexuals suggests that the shift in reproductive mode may not entail a major reorganization of the cell machinery”. In arthropods, the vast majority of genes expressed in female gonads are linked to oocyte differentiation (vitellogenesis etc). There is a very small part in the gonads (the germarium) where meiosis actually happens, and meiosis-related genes have generally very low expression in the germarium of sexual females (and sometimes meiosis actually happens in juvenile stages). This means that there could still be a major reorganization of the cell machinery linked to parthenogenetic reproduction, but such a reorganization could not be detected in the dataset because it would manifest as minor gene expression differences. I suggest omitting or reformulating this statement.

2) Could the authors clarify in the methods whether sexual females were virgin or mated (this has major effects on females in some species, eg *Drosophila melanogaster*), and whether sexual and asexual females were collected while reproductively active? (In many arthropod species, there is a time lag between the moult leading to sexual maturity and the onset of reproduction). For example, little gene expression difference between sexual and asexual females would perhaps be the expected pattern if females were not yet reproductively active.

Author's Response to Decision Letter for (RSPB-2021-1720.R0)

See Appendix B.

Decision letter (RSPB-2021-1720.R1)

31-Aug-2021

Dear Dr Huylmans

I am pleased to inform you that your manuscript entitled "Transitions to asexuality and evolution of gene expression in *Artemia* brine shrimp" has been accepted for publication in Proceedings B.

Data Accessibility section

Open Access

Paper charges

Sincerely,

Professor Gary Carvalho

Associate Editor:

Comments to Author:

Many thanks to the authors for their careful revisions. I very much enjoyed reading the paper!

Best wishes,

Judith Mank

Appendix A

Thank you for the constructive feedback. In light of these comments, we have made several major changes to the manuscript:

1. We now use each of the sexual species as reference for calling sex-biased genes and comparing their expression in asexual lineages. Results using *A. sp. Kazakhstan* as the reference are very similar to those using *A. sinica* as the reference, showing that the patterns we observe are not simply due to a shift in sex-biased expression in the whole Kazakhstan-asexuals cluster. Since we have fewer replicates to call sex-biased genes in *A. sp. Kazakhstan* and *A. urmiana sexual* than in *A. sinica*, we still show the comparisons to *A. sinica* in the manuscript, and summarise comparisons using the others in the supplementary material (Suppl. Table 1, also available here: <https://docs.google.com/spreadsheets/d/1TcTr-wO6lzssy1yWCLUXW32SX4mtVBzBAdG2FpiolAs/edit?usp=sharing>). Results using *A. urmiana sexual* as the reference are inconsistent, but this seems to be due to shifts in this particular lineage, which also shows a much smaller number of sex-biased genes in the gonad, something that we now discuss in the manuscript.
2. We also use each of the sexual lineages as reference for calling differentially expressed genes between sexual and asexual lineages, and have added the respective figures to the supplementary material (suppl. figs. S7 & S8).
3. We removed the ancestry analysis, and instead estimated F_{st} between different lineages, after inferring SNP frequencies directly from the pooled sequencing reads. The resulting plot can be found in figure 1A.
4. We added a PCA - linear discriminant analysis to infer an expression “maleness” score for each individual, which we then apply to asexual individuals. This approach clearly shows that there is no global masculinisation of the transcriptome of asexual individuals, which are consistently predicted by the model to be females. This is now shown in Figure 3C for gonads and suppl. figs. S10 and S11 for heads and thorax, respectively.
5. We have toned down the claim that meiosis genes tend to be differentially expressed, as we agree that this was not well-supported by our data. Additionally, we have checked for the enrichment of known *Drosophila* meiosis genes among our differentially expressed genes and results here are somewhat inconsistent. We now address this in the text (L. 173-185).

Finally, we have made our analyses easily reproducible by providing input files and annotated pipelines in a Git page: <https://git.ist.ac.at/bvicoso/artsexasex> , such that parameters of interest can be further investigated by readers.

We have also answered each of the comments below.

Associate Editor

Comments to Author:

All three reviewers were generally enthusiastic, although there were several important concerns raised about the phylogenetic and other methods.

Reviewer 1 has several questions, most importantly with the strategy of using a single sexual species as a comparison for all parthenogenetic ones for gene expression analyses. Reviewer 2 makes several helpful suggestions for further analyses that will help the authors to make more concrete conclusions.

We address these in detail below.

Reviewer 3 raises a very interesting idea that the turnover of sex biased genes and the masculinization observed in asexual lineages might in fact be a product of regression toward the mean rather than a real biological process, and it would be very helpful if the authors can confirm or refute this. We had a similar issue with a previous analysis (Pointer et al. PLOS Genetics 2013) that might be helpful.

Since we are not measuring sex-bias in the asexuals, regression toward the mean would only be expected if female-biased genes tend to have ancestrally high expression, and therefore lower expression in asexuals (and the opposite for male-biased genes). Review Figures 1 and 2 below shows that while regression to the mean may play a role for the patterns of expression in female heads (although the smaller number of sex-biased genes in this tissues makes it hard to draw strong conclusions), patterns of masculinisation of expression in gonads seem to largely hold independent of the ancestral level of female expression.

This suggests that regression to the mean is unlikely to fully explain the patterns, supporting a potential contribution of the fast expression divergence of genes with sex-related functions (and/or of other unusual features of sex-biased genes).

Review figure 1: Change in ovary expression in different lineages relative to *A. sinica*, as a function of *A. sinica* ovary expression (as a proxy for ancestral expression). Grey dots represent unbiased genes, dark green female-biased genes, and light green male-biased genes. The lines correspond to Lowess-smoothed lines. Boxplots show the distribution of $\text{Log}_2(\text{focal species RPKM} / A. \text{ sinica RPKM})$ for the 25% of genes with the highest *A. sinica* expression; these show patterns largely similar to those observed with all genes (Fig. 3 in the manuscript).

Review figure 2: Change in female head expression in different lineages relative to *A. sinica*, as a function of *A. sinica* female head expression (as a proxy for ancestral expression). Grey dots represent unbiased genes, dark green female-biased genes, and light green male-biased genes. The lines correspond to Lowess-smoothed lines. Boxplots show the distribution of $\text{Log}_2(\text{focal species RPKM} / \text{A. sinica RPKM})$ for the 25% of genes with the highest *A. sinica* expression; these show patterns largely similar to those observed with all genes (Fig. 3 in the manuscript).

I had a few additional questions not raised by the reviewers:

How certain is that the asexual lineages are fully and completely asexual? The Structure plot in Fig 1B calls one of the Atanasovsko populations into question. How was asexuality determined in these lineages?

Cyclical parthenogenesis, in which the same female alternates between sexual and asexual reproduction, is not found in *Artemia*. However, the demographic/genetic boundary between sexual and asexual strains seems to be somewhat permeable. Asexual females occasionally (~1/1000) produce a functional rare male that is capable of fertilising closely related sexual females (at least in the lab), eventually leading to the creation of male hybrids and new hybrid asexual females. Not

only do these new asexual females carry part of the sexual genome, but the hybrid males can themselves further transmit asexual genetic material to the sexual population. It was also recently found that asexual females can be fertilised and produce a minority of offspring sexually when presented with closely related sexual males (Boyer et al, 2021, <https://doi.org/10.1002/evl3.216>). How much of this actually happens in the wild is still unclear, but there is clearly potential (and some evidence) for genetic exchange between sexual and asexual lineages.

Also, with regard to Fig 1B – I assume some species have multiple columns because they were sampled from multiple populations? If so, it might help reduce reader confusion to mention this in the figure legend.

These correspond to our different replicates from the same population. As the reviewers pointed out that these data were not appropriate for an ancestry analysis (as they were pooled), this part of the figure has been removed.

Reviewer(s)' Comments to Author:

Referee: 1

Comments to the Author(s)

This paper addresses gene expression shifts between sexual and parthenogenetic species of *Artemia* brine shrimp, a fascinating system with several closely related sexual and parthenogenetic species. The study is based on a rich dataset and addresses interesting and timely questions, nicely fitting the scope of *ProcB*. However, I have concerns regarding how the analyses and comparisons were done, especially because a single sexual species was used as a comparison for all parthenogenetic ones for gene expression analyses. I understand that choosing the best comparisons is not straightforward in the system given the unresolved phylogenetic relationships among the parthenogenetic lineages and the closest sexual relatives, but I do not see how the topology in Figure 1 justifies the chosen single comparisons. I clarify this (and other) points separately for the different manuscript sections.

Thank you for the supportive comments, and for the suggestions on how to make the analyses more reliable. We have rerun our analyses and they can indeed show that the results are robust. We respond to specific comments below.

Part II: Small shifts in expression associated with asexuality

Here the authors want to identify parallel shifts in gene expression between sexual and parthenogenetic females. They compare each of the 3 parthenogenetic species to a single sexual species (Kazakhstan), call DE genes and search for overlap between the different DE gene lists. They find little (but significant) overlap, which they consider “surprising” given that sex-> asex transitions are sometimes hypothesized to “be accompanied by a major reorganisation of gene expression and molecular and cellular organisation”.

I am not particularly surprised that there is little overlap among the 3 pairwise comparisons, given that only 2 replicates are available in each asexual species. This means that there is little power to detect DE

genes in each comparison, and even less to detect overlap. I suggest the authors screen for parallel shifts independently of cutoffs for significant differential expression.

Thank you for this suggestion. In our original DE analysis, we did not use a fold change cut off but used all genes with adjusted P-values < 0.05. When we use a fold change cut off of 2 instead, we get many more differentially expressed genes but many of them may be biologically meaningless as they have very low expression or large differences between the replicates (see new figure 2 and supplementary figures S9). For this reason, we think that using the adjusted P-value but no fold change cut off makes this analysis more reliable, but now discuss explicitly the caveat of low statistical power:

L145: "Genes that are differentially expressed in at least two of the individual analyses tend to have high fold change differences and very low adjusted P-values (fig. 2), and the low number of shared genes among all three asexual lineages may be due to limited statistical power to detect small changes. However, genes with large expression differences are likely the most biologically relevant, and the fact that so few consistently change in asexuals suggests that the shift in reproductive mode may not entail a major reorganization of the cell machinery. "

L272: "While it is possible that we do not have enough statistical power to detect genes with minor changes, several biological reasons could be behind this small set of core asexuality genes."

When addressing the question of which portion of the transcriptome shifts during the sex-asex transition (how "major" is the reorganisation), I would further be interested in the results of individual comparisons, not the overlap among different shifts.

We have changed figure 2 accordingly to show the volcano plots for individual comparisons instead of the Venn diagrams, which are now found in the supplementary material (suppl. figs. S6-S8). Importantly, both the PCA (Figure 1) and PCA-LDA (fig. 3C and suppl. fig. S10) also support the idea that shifts in expression in asexual species have been limited.

The 3 pairwise comparisons are of course not mutually independent since they all involve the same sexual species (the authors address this point by pooling the asexuals). But I do not understand why the authors do not use different sexual species for comparison, since they are available. If every asexual is compared to all three sexual species for identifying DE genes, the authors could give a range of genes (min-max) with parallel changes, depending on the set of sexual species used for comparison (or at least the two closely sexual species could be used instead of just one).

We agree that this is a good idea and we have done the comparison also with *A. sinica* and *A. urmiana* sexual females. The corresponding Venn diagrammes can be found in the supplementary material (suppl. figs. S7 & S8). We find more DEGs when using *A. urmiana* sexual and even more using *A. sinica*. This, however, is to be expected due to increasing phylogenetic distance. This

confounding factor and the fact that all asexual lineages seem to have originated from an *A. sp. Kazakhstan*-like ancestor made us choose *A. sp. Kazakhstan* females as the primary species of comparison with the asexuals. Ideally, we would have compared species pairs and (based on published phylogenies) we initially expected that at least one asexual lineage would be the direct sister species of *A. urmiana sexual*. However, this is not the case and thus *A. sp. Kazakhstan* seems like the most appropriate control for all comparisons. We have added to the text:

L167: "We also tested for differential expression using the more distantly related sexual species A. urmiana sexual (suppl. fig. S7) and A. sinica (suppl. fig. S8) as reference. As expected, we find more differentially expressed genes with increasing phylogenetic distance, but again only a small proportion of differentially expressed genes are shared between the three asexual lineages. "

Part III: Evolution of sex-biased genes in asexual lineages

In this part of the ms, the authors study the fate of sex-biased genes in sex-asex transitions, again, via pairwise comparisons with only 1 sexual species (which is also used to categorize genes into M-biased, F-biased and unbiased). Surprisingly, they do not use the same sexual species as in the previous part, but a more distantly related, sexual “outgroup” (*A. sinica*). Again, why not use the two closely related sexual species (*Kazakhstan* and *urmiana*) to study the fate of genes that are sex-biased in either one of these species in the three asexuals? This would allow for a discussion of patterns independently of shifts between the outgroup and the clade comprising sexual and asexual species highlighted in grey in fig 1.

We thank the reviewer for this suggestion and had, in fact, also looked at the comparisons to the other two sexual species. We have chosen *A. sinica* as the main species for comparison because we wanted to check for masculinisation of gene expression in the whole clade of Eurasian *Artemia* not only the asexual species to ensure that patterns that we find are indeed due to to asexuality (which our results indicate they are not).

Furthermore, we have only half as many replicates for *A. sp. Kazakhstan* and *A. urmiana sexual* compared to *A. sinica* and thus have better power to detect sex-biased genes in the latter. Thus, we have kept the comparison as it was in the main figure (using *A. sinica* as the baseline), but we have added the comparisons using *A. urmiana sexual* and *A. sp. Kazakhstan* as references in a comprehensive supplementary table (suppl. tab. 1).

The comparison to *A. sp. Kazakhstan* shows consistent results with our main comparison to *A. sinica*. Comparisons involving *A. urmiana sexual*, however, give somewhat mixed results. This may be in part be explained because in this species, we detect much fewer sex-biased genes in gonads (621 MBGs, 285 FBGs) compared to *A. sinica* (1,563 MBGs, 1,335 FBGs) and *A. sp. Kazakhstan* (1,426 MBGs, 998 FBGs). Importantly, the fact that using *A. sp. Kazakhstan* as the reference does not change the patterns shows that the masculinisation of expression that we observe is not a feature of this whole clade.

Part IV. Fast evolution of sex-biased genes in sexual *Artemia*

In this section I am not sure that the finding of faster divergence of expression of female-biased genes is a biological results rather than a technical artifact. Although we know from other studies that there is indeed a (weak) correlation between the rate of sequence and expression evolution, the present finding might be a consequence of the methods used.

The authors quantify gene expression by mapping reads from all species to the sexual outgroup (*A. sinica*). This means that faster evolving genes (such as F-biased genes in the present study) will have a lower mapping probability than rel. conserved genes. Reduced mapping probability would appear as reduced expression of F-biased genes in asexuals (and increased expression divergence) even if it was not the case.

At least as supplemental information, I would like to see the same analyses done with mapping reads from each species to its own transcriptome, and inferring orthologs among species (not just for this section but also for the other analyses in fact).

This is indeed a valid point and we had originally used all the individual transcriptome assemblies and mapped the reads for the respective species to them. However, we only find 8,389 orthologous genes present in all transcriptomes, which limits the power of our analyses. Furthermore, the mapping efficiency of the Eurasian lineages to *A. sinica* was almost the same as to the individual transcriptomes due to the close relatedness. Hence, we decided to use the mapping to *A. sinica* for the main text.

We have now included the results for the gene expression analysis and the expression divergence using the individual transcriptomes in the supplementary material (suppl. figs. S13 & S15). All patterns remain the same, supporting the robustness of our results.

Part I: Phylogeny & evolution of asexuality

This part has some strange methodology. First, why would the authors solely use “head samples” for calling SNPs?. Since each sample is a pool of 5 individuals (presumably different ones?), I would pool all samples per species to call SNPs, as this would give the most accurate estimate of allele frequencies in a given lineage.

For most lineages, the same individuals were used for the head, gonads and, when available, thorax sequencing. Pooling would therefore not increase the number of individuals sequenced. We chose heads because they tend to have very conserved gene expression. We feared gonads would be more heterogeneous, and preferred to use the same tissues and numbers of individuals per strain to avoid creating biases in the analysis.

We have however obtained below an Fst figure from combined head and gonad reads (see review figure 3 below). As expected given that we simply have more coverage for the same 10 individuals per strain, this does not change the patterns, with Fst increasing with phylogenetic inputted distance. We therefore have kept the figure based on heads only in the main manuscript.

Review Figure 3: Mean pairwise Fst estimated from combined head and gonad reads.

After calling SNPs in each sample, the authors run the program structure to find genetic clusters. However, structure treats samples as individuals (with 0/0 1/1 or 1/1 genotypes), not taking into account allele frequencies at each locus. I would completely change this section and use a pool-seq analytical approach based on allele frequencies rather than genotypes.

Thank you for pointing this out. Since the only purpose of the analysis was to emphasise the close proximity of *A. sp. Kazakhstan* and of the asexual lineages, we now simply estimate Fst between all population pairs, after inferring allele frequencies directly from the RNA-seq reads (both head replicates, pooled). This is now shown in figure 1, using the head data.

Other comments.

-There is insufficient and sometimes confusing information about the species and biological material used. For example, how genetically diverse are the strains given they were obtained from some type of “stock center”? I was surprised to see so much assignment variation among different pools of individuals from a single strain (Figure 1 B, Suppl. Fig. 2) – especially for the asexual strains.

We apologise, the term “stock center” is indeed somewhat misleading. In the case of *Artemia*, the stock center preserves cysts that were obtained as F1 of the original collected populations, not live colonies; these F1 cysts are kept in the -80 until they are needed. For this reason, we expect that the individuals we obtained for our RNA-seq, which were collected from the first generation to emerge in our lab (i.e. still the F1 of wild-caught animals), to largely reflect the natural diversity of the wild

strains. *A. Aibi Lake* individuals are the exception, since our original samples were not good enough, and new individuals were collected later on for sequencing. We have now clarified this in the methods:

LI. 342:

“At least two biological replicates (using different individuals) were collected per sex and tissue from the first generation to emerge from the cysts (suppl. tab. 2). The RNA-seq samples originally obtained for A. Aibi Lake were used for transcriptome assembly but were outliers in the expression analyses. Hence, we sampled and sequenced two new replicates of each tissue (samples 101424-101429 and 101440-101441, suppl. tab 2), which were used for all downstream expression analyses.”

Estimates of synonymous divergence (using SNPgenie) for each of our head samples of 5 pooled individuals are provided below (Review Figure 4). Most samples have a median PiS around 0.01, and a mean PiS of around 0.015. Three samples stand out: aib_40773 and aib_40774, which were excluded from the expression analysis because of their low mapping rates, and for which we have low power to call SNPs, have unusually low diversity. Ata_45191 shows the highest diversity, as pointed out by the reviewer.

Review figure 4: Distribution of PiS for all head samples. PiS values were estimated using SNP genie from the same VCF file that was used for the Fst calculation and is described in the methods. The full

summary of how these values were obtained is available here:

<https://git.ist.ac.at/bvicoso/artsexasex/-/blob/master/Extra/SNPgenie.md> .

In some cases (notably Aata) I was also wondering if the authors verified that these were in fact diploid and not polyploid? Why did the authors not use the transcriptome data from “thorax” indicated in the methods?

While we did not directly assess ploidy, we have multiple lines of evidence that the *Artemia* lineages used for this study are diploid rather than polyploid. First of all, the BUSCO score in all transcriptome assemblies gives low values of duplication, comparable with other diploid Arthropod de novo assemblies. In a polyploid, we would expect to see elevated rates of duplication. In addition, we observe rare males in some of our parthenogenetic lineages (particularly common in populations derived from the asexuals from Aibi lake) which have only been described in diploid *Artemia*. Finally, the pi values for synonymous substitutions also point towards diploidy as in polyploids generally higher levels of diversity would be expected. The one exception is the unusually polymorphic sample of *A. Atanasovsko* (the only population where both diploid and polyploid asexuals have been found). However, since patterns of expression were not unusual for this sample, even if there is contamination with a polyploid in this sample this does not seem to interfere with our results.

Why did the authors not use the transcriptome data from “thorax” indicated in the methods?

We had some difficulty growing enough *A. urmiana sexual* individuals for the dissections, and could not get enough thorax tissue. Since thorax data is missing for this species, and produces results largely similar to heads (as expected from another somatic tissue), we decided to focus on heads and gonads in the main text. Corresponding figures for thorax have been added as supplementary figures S4-5 & S11.

-The authors state that homologs of genes differently expressed between sexual and asexual females in both heads and gonads “are all involved in meiosis and/or cell division [2], suggesting that they may play a role in the modified meiosis required for asexual reproduction in brine shrimp”. I think this is a bit of an over-interpretation since there are no meiotic divisions in the head.

We thank the reviewer for this comment and agree that we were overstating the influence of meiosis genes. We have now changed the text and in addition have checked for the presence of known *Drosophila melanogaster* meiosis genes among our differentially expressed genes. The results are somewhat inconsistent and we address this now:

LI. 177:

“To systematically test for enrichment of meiosis genes, we searched for homologs of annotated meiosis genes from *Drosophila melanogaster* [41] in our *A. sinica* transcriptome, yielding 873 putative

homologs. There is no excessive overlap with gonad asexuality genes when asexuals are compared to A. sp. Kazakhstan. When they are compared to A. urmiana sexual or A. sinica we find significant enrichment of meiosis genes among the differentially expressed genes in gonads (13 in both comparisons, $P=4.7e-9$ and $P=7.6e-5$, respectively, FET) and heads (6 and 23 genes, respectively, $P=0.006$ and $P<2.2e-16$, respectively, FET). These results therefore hint at, but do not fully confirm, a function in meiosis for some of these putative asexuality genes.”

Referee: 2

Comments to the Author(s)

This manuscript looks at patterns of gene expression in male and female brine shrimp of sexual species as well as the gene expression in closely related asexual lineages. One aim is to identify genes with expression patterns that change in a consistent way during the transition to asexuality, which could be informative about how this transition occurs functionally, and the authors find some genes that seem to show consistent changes. Another interesting aim which the Introduction spends quite a bit of time on is to test ideas about how relaxation of sex-specific selective pressures alters gene expression patterns during the transition to asexuality, but the results here are a bit difficult to interpret, although that may be able to be addressed.

Thank you for the helpful suggestions, which we address below. We agree that the results are complex, but hope that you will find our efforts to disentangle the effect of asexuality from those of general properties of sex-biased genes useful for the field.

My comments below.

- For the PC analysis, do other PCs show anything interesting? For example, in Fig. 1, do heads separate by sex on one of the later PCs? There is plenty of variance still to explain after 1/2. If there is an axis that seems to separate male and female samples in the sexual lineages, and I think it would be weird if there wasn't, you may find an interesting/informative pattern in how the asexuals show up on that PC.

This is a good point, and we agree that PC1 and PC2 do not tell the whole story. In heads, no single PC was able to clearly separate males and females (we stopped at PC10). Following your suggestion (see below), we ran a PCA-LDA analysis on the data to obtain a “maleness score” for each individual. The resulting model assigned 100% of asexuals as females using gonad and 83% using head data, and showed no evidence of global masculinisation of expression. We now show this in figures 3C for gonads and in suppl figs. S10-S11 for heads and thorax.

- On line 174, what are the chance expectations? And report the stats here for the comparison between observed and expected. This is actually a fairly important statement and the support needs to be shared.

We report the numbers of expected genes together with the test and the P-values in the sentences following this statement:

L 162: “Namely, we find four genes up-regulated and 26 genes down-regulated in heads (versus 0.05 and 0.1 expected, $P=4.7e-07$ and $P< 2.2e-16$, SuperExactTest). For gonads, nine genes are up- and 27 are down-regulated in all three comparisons (versus 0.5 and 0.2 expected, $P< 2.2e-16$ for both, SuperExactTest). 48 of these 59 genes were also identified in the combined analysis (“*” in suppl. tabs. 5 & 6).”

- l. 196-200 – At least as written currently, I think this is a pretty tenuous connection. Just from a quick look, the specific gene mentioned here seems to have diverse functions outside of meiosis. It may be that there is a stronger link to be made between meiosis and this gene (and for the 21 out of 68 mentioned previously), but as I read the text now that link does not seem convincing.

We thank the reviewer for this and agree that this part of the manuscript was not written very convincingly. We have now completely changed the text. Please also see the general response at the response to the editor (point 5).

- Most of my comments are related to Part III, which to me is the most important/interesting section in light of the setup of the paper. After thinking about this section for a while, particularly the concluding sentences, isn't the main conclusion here that all the species along the branch you're interested in using to test these ideas about sexuality (2 species) versus asexuality (3 species) are in fact showing less sex-biased gene expression than the outgroup, and are fairly similar in masculinity/femininity of gene expression? If that is the case it's not a particularly satisfying test of the idea, because as the authors write here it could just be a feature of the Eurasian artemia. Then we don't know whether or not the lack of masculinization or feminization is due to a lack of the putative selective forces that would cause that or instead (for example) a lack of time for masculinization or feminization to take place...

The reviewer is right that we find more sex-biased genes in *A. sinica* than in the other two sexual species (see numbers in tab. 1 in the supplementary results). However, especially for *A. sp. Kazakhstan*, this difference is not very strong, and is presumably due to reduced statistical power because we have twice as many replicates for *A. sinica* than for the other two sexual species.

Following yours and Reviewer 1's suggestions, we have added similar comparisons, but using *A. urmiana sexual* and *A. sp. Kazakhstan* as references, in a comprehensive supplementary table (suppl. tab. 1). While using *A. urmiana sexual* as the baseline gives mixed results, patterns using *A. sp. Kazakhstan* as the reference are consistent with those using *A. sinica*, showing that they do not reflect a general feature of Eurasian *Artemia*. We now explicitly mention this:

Ll. 226:

“Finally, similar patterns of “masculinisation” are observed when *A. sp. Kazakhstan* is used as the reference to call sex-biased genes and as the proxy for ancestral expression (suppl. tab. 1), further arguing against a shift in expression in the whole *Kazakhstan* clade. Using *A. urmiana sexual* as the

reference gives inconsistent results (suppl. tab. 1), although this seems to be due to peculiarities of the data or biology for this species, which also yields much fewer sex-biased genes in the gonad.“

Related to my earlier comment about the PC visualization, it seems to me like a better way to test the idea in the Introduction—do asexuals look feminized or masculinized—would be to approach this in a multivariate way. For example, you could do a PCA on a sexual lineage, or both (e.g. urm, or both urm and kaz), males and females, which will give you a clear separation of males and females on PC1. Then plot the asexuals on the same PC to see if they fall out as intermediate, more male-like, more female-like, ‘uber’ female, ‘uber’ male, etc. This could be done for each tissue and is a direct test of whether the asexuals are feminized or masculinized or neither. It allows you to pool information about the overall profile of maleness and femaleness, rather than looking gene by gene, and also allows you to be more quantitative about it, because it assigns each of the samples a numeric masculinity or femininity.

We thank the reviewer for this helpful suggestion and we have now included a PCA linear discriminant analysis. For gonads, this can be found in the main figure 3 in panel C and the other tissues can be found in the supplementary material (S10 and S11). Asexuals are consistently placed with the females (except in thorax, where the small number of sexual samples did not seem to be sufficient to train the model), showing that there is no overall masculinisation of gene expression. We decided to continue to show the box plots in figure 3 so that it is easy to compare our results to the findings in *Timema* stick insects (Parker et al. 2019).

Maybe it would be better to use discriminant analysis, actually, to specifically define male vs. female in urmiana or urmiana+kazakhstan, and then plot your other species on the discriminant axes. Right now as I understand it we are limited to interpretation of Figure 3, which says that they’re all kinda sorta going the same way relative to *A. sinica*.

As stated in response to the previous comment, we have run a PCA-LDA analysis and the results are shown in figure 3C.

Also, given the discussion of the observed rapid turnover of sex-biased genes in other groups, why use *A. sinica* to define sex-biased genes? According to the phylogeny, it seems like you should be able to use *A. urmiana* to define the genes, and the result would be a much more consistent pattern of a male- or female-biased gene where the other species you’re interested in much more consistently show the same bias.

We have tried all sexual species as the baseline to define sex-biased genes (suppl. Tab. 1) as well as using the overlap between all three (i.e. the genes that conserved their sex-biased expression, S12). Results are consistent for when either *A. sp. Kazakhstan* or *A. sinica* are used to define sex-biased genes and ancestral expression, but we seem to have reduced statistical power for *A. urmiana* sexual (see answer above for comment 4).

• I don’t see any mention of this, but it is probably worth discussing whether allometry could impact the results—are the tissues being profiled wildly divergent in sizes, or the sizes of subcomponents?

Since we are using single tissues, only allometry changes of subcomponents/cell types could affect the results (as RNA-seq only gives relative measures of expression, such that the amount of starting material/tissue size is not relevant). For our purposes, it does not matter whether changes in expression are driven by changes in transcription within each cell, or by changes in cell composition, since we do not make assumptions or claims about the mechanism. That said, as far as we know no major morphological differences have been described in this group (but it is not clear that this has been studied systematically in Eurasian *Artemia*).

- L. 258-289 says, “If the rate of turnover of sex-biased genes in the Eurasian *Artemia* species is very high, these genes should show signatures of fast evolution.” Is that true? Wouldn't it be more likely actually that the turnover is caused by regulatory regions outside of the genes, and that is where you would see the sequence-level changes?

Sex-biased genes have been shown to evolve fast, both at sequence and expression level. While it is not completely clear why this is, it is likely that similar selective pressures are leading to the fast evolution of both the coding and regulatory sequences (e.g. strong sexual selection, or reduced pleiotropy of genes with gonad-specific expression). However, we agree that expression divergence is the more relevant measure when discussing the turnover of sex-biased genes, and have now moved the dN/dS analysis to the supplementary materials.

Also, don't sex-biased genes in general, across taxonomic groups, show signatures of fast evolution? In that case you would then expect to see this signal that confirms a widespread pattern regardless of the specific rates of turnover occurring here.

We completely agree that fast evolution is commonly seen for sex biased genes of various species (and now mention this explicitly in the text). The quick change in sex-specific expression is, as far as we know, typically understood to be the cause of turnover in sex-biased genes [44,45], such that the fast evolution at the expression level does seem to suggest a high rate of turnover (again, something in line with what has been observed in other species).

- Some smaller things:

- l. 140 – “We head...” should be “We profiled head...”

Corrected.

The figure 1 legend needs to be elaborated for clarity and written less informally. Also principle should be principal.

This has been fixed.

l. 178-179 – I’m a little skeptical of $1e-53$ and $1e-18$ p values in this test... How do we interpret $1e-53$, really? It seems likely that the statistical test is inappropriate.

We agree that the numbers here are very low for such small numbers of genes in the overlap. However, we have specifically chosen this test because it is designed to test for significance in multi-set intersections and the numbers of transcripts in the whole analysis is very large. To hopefully convince the reviewer that these low P-values are indeed realistic, we have tested the overlaps between each 2 species with Fisher’s Exact Test (exemplified for genes up-regulated in heads where we found the very low P-value):

Aata vs. Aurm: P-value = $3.178e-07$

Aata vs. Apar: P-value $< 2.2e-16$

Apar vs. Aurm: P-value $< 2.2e-16$

The SuperExactTest calculates the P-value even if it is below $2.2e-16$ and thus, we had reported it this way. We have now changed it to the R convention and simply report it as $< 2.2e-16$.

Same problem with figure 2 legend, too informal. E.g. “A) shows up-regulated and B) down-regulated genes in asexuals heads.”

Figure 2 has been replaced and thus the figure legend has changed.

Figure 3 also needs work. For example, the legend begins with “Gene expression in...” but this figure shows relative gene expression. It would also be nice to have something clarifying A is females and B is males without having to read the legend (for example, male and female symbols or the words male and female above the panels). And the y axis is hard to parse, actually, maybe there is some way to make this more readable while also making it clear that every single plot here is a comparison to *A. sinica* counterparts?

The figure legend has been changed, the size of the font on the y-axis has been increased and the term “male”/“female” therein has been replaced by the same terms above the plots at A) and B).

Referee: 3

Comments to the Author(s)

In this article, Huylmans and colleagues measure gene expression in 3 sexual and 3 asexual lineages (or species) in *Artemia* brine shrimps, to study how gene expression is affected by transitions to asexuality. They identify genes systematically less or more expressed in asexuals than in sexuals, and show that meiosis genes are overrepresented in this list. They also test whether the release of sexual conflict between males and females (sexual conflicts do not occur anymore in asexual lineages) affects the evolution of gene expression.

Main comments:

Meiosis genes: The authors show that meiosis genes are common in the set of genes differentially expressed between sexual and asexual as 21 out of 68 genes have a meiosis-related function. But they do not explain how they determine that a gene has a meiosis-related function, and the 21 genes are not identified in the supplementary tables. The authors should also demonstrate statistically that 21 out of 68 is more than expected (thus measure the number of genes with meiosis-related functions in the list of genes that are not DE between sexuals and asexuals).

We agree that this section was not fully supported and have revised it substantially. As already pointed out in response to the other reviewers and in the general comments, we have deleted the blast result and added a comparison to known *Drosophila* meiosis genes instead. The results are somewhat inconsistent and thus no real conclusion as to the function of the asexuality genes can be drawn.

Masculinization of expression: I wonder if the fact that female-biased genes (identified in a sexual species A) tend to be systematically less female-biased in the other species is not an expected consequence of the way the analyses are done. Let's suppose that with RNAseq we just get an estimate of the true level of expression for a gene (so this estimate is of course imperfect). Hence, those genes identified as female-biased in species A are those genes that were the most female-biased in the noisy RNAseq data (female-bias is due to true female-biased expression effect but also to some stochastic effects that may have increase the bias). So, by construction, when looking at the expression of female-biased genes defined in species A in another species, in most cases these genes will be less female-biased. I have no idea if the effect is tiny or not, but I think it adds to the other mechanisms hypothesized by the authors.

This is a very good point. We have tested for a possible effect of regression to the mean (see our response to the editor, and review figs. 1 and 2). In particular, we look at expression changes of different classes of sex-biased genes across different levels of ancestral expression. While we find that there is a slight effect in female expression in heads, patterns for gonads, which are shown in the main text, seem to hold independent of ancestral expression.

Origin of samples: More information on the geographical origin and areas of distribution of the different species should be provided.

We have now added a paragraph to the supplementary methods with the information available on the origin of the different species/populations:

1. *Diploid parthenogenetic (A. parthenogenetica) from Urmia Lake (West Azerbaijan, Iran), 37°21'17"N - 45°38'45"E. Lab obtained cysts during 2009 from individuals hatched from a sample collected in the wild on January 2009.*
2. *Diploid parthenogenetic (A. parthenogenetica) from Aibi Lake (Xinjiang Province, PR China), 44°53'05"N - 82°53'55"E. Lab obtained cysts during 2011 from individuals hatched from a sample collected in the wild on August 1991.*
3. *Diploid parthenogenetic (A. parthenogenetica) from Atanasovsko Lake (Bulgaria), 42°29'39"N - 27°25'54"E. Original cyst sample collected in the wild on October 2006.*

4. *Sexual (A. sinica) from Tanggu Salterns (Tianjin Province, PR China), 38°55'00"N - 117°37'00"E. Lab obtained cysts during 2006 from individuals hatched from a sample collected in the wild on September 2005.*
5. *Sexual (A. sp) from an unknown locality in Kazakhstan. Lab obtained cysts during 2010 from individuals hatched from a sample collected in the wild in an unknown date. This is the sample #1039 of the Artemia Reference Center (Ghent University). It has been widely studied in phylogeography, evolution, etc. investigations by different researchers, but whose origin is unknown. It was provided by the Dutch company CATVIS, S.V. to the ARC.*
6. *Sexual (A. urmiana) from Urmia Lake (West Azerbaijan, Iran), 37°21'17"N - 45°38'45"E. Lab obtained cysts during 2012 from individuals hatched from a sample collected in the wild on January 2009.*

Phylogeny: No branch support is provided for the phylogeny (Figure 1A). Such information is important especially as the authors compare the clustering of species to a previous study (lines 121-122).

All branches have 100% bootstrap support. We now mention this in the figure caption.

RNAseq samples: It is not clear if a single clone was used for asexual species or if different clones were pooled. Similarly, it is also not clear if RNAseq replicates correspond to the same pool of 5 individuals (sequenced twice) or to 2 pools of 5 different individuals.

We apologise that this was not written clearly enough. We now explicitly state this in the methods section:

L342: "At least two biological replicates (using different individuals) were collected per sex and tissue from the first generation to emerge from the cysts (suppl. tab. 2)."

Admixture analyses of the RNAseq samples: This method is not adapted to the data for different reasons. The admixture analysis has been developed in a population genetic framework. This analysis clusters samples (individuals) into populations by minimizing Hardy-Weinberg disequilibrium within the inferred populations (hence per population allelic frequencies are estimated) and assuming linkage equilibrium. Thus many samples (individuals) originating from different populations are required. This is clearly not the case with the data analyzed here (a few samples per species). Furthermore, asexual lineages usually do not satisfy HW assumptions and linkage equilibrium. A possibility to look at the relationship between samples would be to construct a tree based on population genetic distances between samples.

We agree with the reviewer that this analysis was not appropriate. We have removed it from the manuscript and have instead added an Fst analysis (see fig 1 and general response above).

Furthermore, since there is no precise description of the content of the RNAseq samples, I have additional questions. I suppose that each sample (for sexual species) is a mix of 5 genetically different individuals, but then each sample is analyzed as a diploid sample for SNPs calling. This is weird. Are the two

replicates for a tissue sequencing replicates or do the individuals used differs? (this will affect the similarity of the replicates).

The individuals that were used differ. We now infer allele frequencies directly from the number of reads supporting the reference and alternative alleles for each population (combining the two replicates, i.e. using 10 individuals / population) for a pooled Fst analysis.

Lines 50-51: "..., while other studies found much more subtle differences in expression between sexual and asexual lineages [25-27]." None of the 3 cited articles are about gene expression.

You are completely right. We have changed the text in this paragraph and are more explicit about the current studies looking at changes between sexuals and asexuals and the implications for gene expression:

LI. 41:

"Comparisons of gene expression have been used to investigate both the molecular basis and the consequences of asexuality [16,23,24], and detected many hundreds of genes differentially expressed between sexual and asexual morphs/lineages. Similarly, Duncan et al [25] found evidence of an entirely different developmental programme underlying the asexual part of the life cycle of the pea aphid. On the other hand, single loci often seem to control the shift between sexual and asexual states [26-28]. Whether this simple genetic architecture translates into large or subtle changes in gene expression is still unknown."

Minor comments:

The aim of the study in the introduction is vague (line 101: "... allowing us to test some of the questions on the evolution of asexuality.")

We have specified our questions better:

L89: "[...] allowing us to test whether a core set of genes changes consistently with the evolution of asexuality, and whether we can detect a feminisation in expression patterns in asexual females, consistent with a release from sexual antagonism."

Two A. parthenogenetica samples were removed. Could you mark them on the suppl fig 3 and 4 (I see more than two A. parth samples that do not cluster so well).

The excluded samples have been marked. We have also rerun our comparison of focal/ancestral expression with further samples removed (for each tissue/species, we used the two most correlated samples as the reference, and removed the few samples with a Spearman rank correlation below 0.75 to these "good" samples), but this did not change any of the patterns (Sup. Table 1).

Line 258: Why expect fast evolution if rate of turnover a sex-biased gene is high?

Genes with sex-biased expression often show high rates of divergence at both the sequence and expression level. Two (non mutually exclusive) hypotheses have been extensively put forward to explain this:

- 1. Genes with sex-biased expression (especially male-biased ones) tend to be fairly tissue-specific, such that they are under fewer pleiotropic and therefore selective constraints;**
- 2. Sexual selection may tend to target genes with sex-specific functions.**

Independent of what drives this fast evolution, changes in expression of a sex-biased gene can either exacerbate the sex-bias or remove it (ie lead to turnover), and the latter seems to happen frequently. Since there is a more direct connection between changes in expression and turnover, we have removed the dN/dS plots from the main manuscript.

Appendix B

Associate Editor

Comments to Author:

One of the original reviewers and myself have read through the revised manuscript, and are very happy with the new version. I would like to thank the authors for their careful responses to the suggestions in the previous round of review, and their thorough revision. The reviewer has two minor comments that should be very easy to deal with, and I do not expect to send a revised version back to review.

Thank you! We have made the two changes requested below.

Reviewer(s)' Comments to Author:

Referee: 1

Comments to the Author(s).

I think the authors have promptly addressed all comments raised on the previous version. I only have two more minor comments.

- 1) In L148-150, the authors argue that “genes with large expression differences are likely the most biologically relevant, and the fact that so few consistently change in asexuals suggests that the shift in reproductive mode may not entail a major reorganization of the cell machinery”. In arthropods, the vast majority of genes expressed in female gonads are linked to oocyte differentiation (vitellogenesis etc). There is a very small part in the gonads (the germarium) where meiosis actually happens, and meiosis-related genes have generally very low expression in the germarium of sexual females (and sometimes meiosis actually happens in juvenile stages). This means that there could still be a major reorganization of the cell machinery linked to parthenogenetic reproduction, but such a reorganization could not be detected in the dataset because it would manifest as minor gene expression differences. I suggest omitting or reformulating this statement.

This has been changed to (L.148 in the tracked pdf):

[...] the low number of shared genes among all three asexual lineages may be due to limited statistical power to detect small changes (on the other hand, genes with large expression differences are likely the most biologically relevant).

- 2) Could the authors **clarify in the methods whether sexual females were virgin or mated** (this has major effects on females in some species, eg *Drosophila melanogaster*), and whether sexual and asexual females were collected while reproductively active? (In many arthropod species, there is a time lag between the moult leading to sexual maturity and the onset of reproduction). For example, little gene expression difference between sexual and asexual females would perhaps be the expected pattern if females were not yet reproductively active.

We now give this information (L.338 in the tracked pdf):

Virgin adults were maintained at 60g/l salinity (asexuals produced offspring), and dissected to obtain head and gonad tissue (all species), thorax (all but *A. urmiana* sexual) and whole bodies (*A. sinica*, *A. sp. Kazakhstan*, *A. Aibi Lake*).